# Learning in Multi-Stage Decentralized Matching Markets

**Xiaowu Dai**
UC Berkeley
xwdai@berkeley.edu

**Michael I. Jordan**
UC Berkeley
jordan@cs.berkeley.edu

## Abstract

Matching markets are often organized in a multi-stage and decentralized manner. Moreover, participants in real-world matching markets often have uncertain preferences. This article develops a framework for learning optimal strategies in such settings, based on a nonparametric statistical approach and variational analysis. We propose an efficient algorithm, built upon concepts of "lower uncertainty bound" and "calibrated decentralized matching," for maximizing the participants' expected payoff. We show that there exists a welfare-versus-fairness trade-off that is characterized by the uncertainty level of acceptance. Participants will strategically act in favor of a low uncertainty level to reduce competition and increase expected payoff. We prove that participants can be better off with multi-stage matching compared to single-stage matching. We demonstrate aspects of the theoretical predictions through simulations and an experiment using real data from college admissions.

## 1 Introduction

Two-sided matching markets have played an important role in microeconomics for several decades [34]. Matching markets are used to allocate indivisible "goods" to multiple decision-making agents based on mutual compatibility as assessed via sets of preferences. Such a market does not clear through prices. For example, a student applicant cannot simply demand the college she prefers but must also be chosen by the college. Matching markets are often organized in a decentralized way. Each agent makes their decision independently of others' decisions, and each agent can have multiple stages of interactions with the other side of the market. College admissions with waiting lists and academic job markets are notable examples. We refer to such markets as *multi-stage decentralized matching markets*.

Uncertain preference is ubiquitous in multi-stage decentralized matching markets. For instance, colleges competing for students lack information on students' preferences. An admitted student may receive offers from other colleges. She needs to accept one or reject all offers within a short period during each stage of early, regular, and waiting-list admissions [5]. This admission process provides little opportunity for colleges to learn students' preferences, which are uncertain due to competition among colleges and variability in the relative popularity of colleges over time. Such uncertain preferences pose a challenge for colleges in their attempt to formulate an optimal admission strategy. Consequently, colleges may end up enrolling too many or too few students relative to their capacity or having enrolled students overly far from the attainable optimum in quality.

This paper addresses the following two research questions: (i) Given the uncertain preferences on one side of the market (e.g., students), how can agents (e.g., colleges) learn an optimal strategy that maximizes expected payoffs based on historical data? (ii) What are the fundamental implications of multi-stage decentralized matching on the welfare and fairness for both sides of the market? We study these two questions using *nonparametric statistical methodology* and *variational analysis*. We propose a new algorithm for maximizing agents' expected payoffs that is based on learning

stage-wise optimal strategies and calibrating state parameters based on historical data. In particular, our algorithm balances the opportunity cost and the penalty for exceeding the quota for calibration. Based on the calibrated state, the algorithm efficiently learns an optimal strategy using statistical machine learning methods. The statistical model not only provides a foundation for the algorithm but it also provides an analytical framework for understanding the implications of the approach for welfare and fairness. We show that agents will favor arms with realistic and stable opportunities for matching instead of only targeting the top-ranked arms. Moreover, we show that agents are better off with multi-stage decentralized matching as compared to single-stage decentralized matching.

Adopting literature from the bandit literature, our model has a set of *agents*, each with limited capacity, and a set of *arms*. Each agent values two attributes of an arm: a "score" that is common to all agents and a "fit" that is agent-specific and independent across agents. Agents rank arms according to their scores and fits. An agent's strategy consists of how many and which arms to pull at each stage. On the other hand, there is no restriction on the preferences of arms. The model allows uncertainty in the preferences, which is incorporated into the arms' stage-wise acceptance probabilities. The acceptance probability depends on the unknown state of the world and the competition of agents at each stage. We consider a simple timeline for multi-stage markets. At each stage, agents simultaneously pull sets of arms. Each arm accepts at most one of the agents that pulled it. The arms have to make irreversible decisions at each stage without knowing which other agents might select them in later stages.

**Our contributions**    There are two main contributions in this paper, which correspond to the two questions above. Our first contribution is to propose a new algorithm that maximizes the agent's expected payoff in multi-stage decentralized matching markets. The algorithm sequentially learns the optimal strategy at each stage and is built upon notions of *lower uncertainty bound* (LUB) and *calibrated decentralized matching* (CDM). The key idea is to calibrate the state parameter in a data-driven approach and take the opportunity cost and penalty for exceeding the quota into account. The calibration can be performed under both average-case and worst-case metrics, depending on whether we are maximizing the averaged or minimal expected payoff with respect to the uncertain state. Given the calibrated state, the algorithm efficiently learns the optimal strategy using historical data via statistical machine learning methods.

The second contribution is providing an analytical framework for understanding the welfare and fairness implications. We show that agents favor arms with low uncertainty in levels of acceptance, suggesting that agents prefer arms with a realistic and stable chance for matching instead of only targeting the top-ranked arms. Such strategic behavior improves the agent's expected payoff since otherwise, by the time that arms have rejected that agent, the next-best arms that the agent has in mind may already have accepted other agents. However, the strategic behavior leads to unfair outcomes for arms because some arms are not pulled by their favorite agents even though these agents pull arms ranked below them. We prove that agents are better off in multi-stage decentralized matching markets compared to single-stage decentralized matching markets.

**Related work**    This paper is related to three strands of literature. The first line is on matching markets. Most theoretical work on matching markets traces back to [21] that formulated a model of two-sided matching without side payments, and [37] that formulated a model of two-sided matching with side payments. The model in [37] is also related to the maximum weighted bipartite matching and its to stochastic and online generalizations [27]. Our goal is to design algorithms for maximizing the agent's welfare under the model of [21], given the uncertain preferences of arms. This is different from the goal of finding a matching with the largest size in maximum matching literature [37, 27]. The second strand of literature is on the decentralized interactions in matching markets [15–17, 30, 35] and search literature [28, 31]. Our paper contributes to this strand of literature via its analysis of multi-stage markets that allow uncertain preferences. We also study the economic implications for strategic behaviors in multi-stage decentralized markets. The third related body of literature is on algorithmic studies of college admissions. The celebrated work in [21] introduced the deferred acceptance algorithm implemented under central clearinghouses. Recent works have been focused on equilibrium admissions, students' efforts, and students' information acquisition costs in forming preferences; see, [7, 11–13, 18, 20, 22, 24]. In contrast, we emphasize students' multidimensional abilities and multiple colleges competing for students. The students' preferences are uncertain due to the competition among colleges and variability in the relative popularity of colleges over time. We develop a statistical model for learning the optimal strategies using historical data.

## 2    Problem Formulation

**Multi-stage decentralized matching markets**    Let $\mathcal{P} = \{P_1, P_2, \ldots, P_m\}$ be a set of $m$ agents. Let $\mathcal{A} = \{A_1, A_2, \ldots, A_n\}$ the a set of $n$ arms. Here $\mathcal{P}$ and $\mathcal{A}$ are the sets of participants on the two sides of the matching market. Each agent $P_i$ has a quota $q_i \geq 1$. We assume that $q_1 + q_2 + \cdots + q_m \leq n$. There are total of $K \geq 1$ stages of the matching process. At each stage, an agent who has not used up its quota can pull available arms in the market. When multiple agents select the same arm, only one agent can successfully pull the arm according to the arm's preference. We denote $[m] \equiv \{1, \ldots, m\}, [n] \equiv \{1, \ldots, n\}$, and $[K] \equiv \{1, \ldots, K\}$. Decentralized matching markets require participants to make their decisions independently of others' decisions [33, 35]. Notable examples of such markets include college admissions in the United States, Korea, and Japan, where $\mathcal{P}$ and $\mathcal{A}$ represent the sets of colleges and students, respectively [5, 6]. Our goal is to learn the agent's optimal strategy for maximizing the expected payoff. A strategy consists of deciding how many and which arms to pull at each stage. Agent's decision-making in decentralized markets faces incomplete information about other agents' decisions and arms' preferences.

**Participants' preferences**    The agents' preferences are based on the arms' latent utilities. Consider the following latent utility model:

$$U_i(A_j) = v_j + e_{ij}, \quad \forall i \in [m], j \in [n], \tag{1}$$

where $v_j \in [0,1]$ is arm $A_j$'s *systematic score* considered by all agents, and $e_{ij} \in [0,1]$ is an agent-specific *idiosyncratic fit* considered only by agent $P_i$, $i \in [m]$. A utility model with a similar separable structure has been widely used in the matching market literature [4, 13, 15].

The arms' preferences have no restrictions and can involve uncertainty. From an agent's perspective, arms accept offers with probabilities dependent on opponents' strategies and arms' preferences. Let the parameter $s_{i,k} \in [0,1]$ be the *state of the world* [36] for agent $P_i$, such that the probability that an arm $A_j$ accepts $P_i$ at stage $k$ is $\pi_{i,k}(s_{i,k}, v_j), \forall i \in [m], j \in [n], k \in [K]$. Since agents compete for arms with a higher score, the acceptance probability $\pi_{i,k}(s_{i,k}, v_j)$ models the agents' competition through the dependence on the score $v_j$. Moreover, $\pi_{i,k}(s_{i,k}, v_j)$ incorporates the arm's uncertain preference into the state $s_{i,k}$. It is known that there exists a valid probability mass function $\pi_{i,k}(s_{i,k}, v_j)$ [15]. We assume that $\pi_{i,k}(s_{i,k}, v_j)$ is strictly increasing and continuous in $s_{i,k}$. Thus, a larger value of the state $s_{i,k}$ corresponds to the case that agent $P_i$ is more popular. In practice, the true state is unknown a priori to $P_i$ and needs to be estimated from data. For instance, the *yield* in college admissions is defined as the rate at which a college's admitted students accept the offers. However, the yield is unknown a priori to the college in the current year [13]. Colleges can only estimate the distribution of the yield from historical data. In this paper, we study a nonparametric model of $\pi_{i,k}(\cdot, \cdot)$ by assuming it belongs to a reproducing kernel Hilbert space (RKHS) [3, 39]. Later, in Section 3.2, we propose an algorithm for calibrating $s_{i,k}$ and efficiently estimating $\pi_{i,k}(\cdot, \cdot)$ using historical data. Given the latent utility $U_i(A_j)$ and the acceptance probability $\pi_{i,k}(s_{i,k}, v_j)$, agent $P_i$'s expected utility of pulling arm $A_j$ at stage $k$ is $(v_j + e_{ij})\pi_{i,k}(s_{i,k}, v_j)$.

**Timeline of the matching**    First, Nature draws a state such that arms' preferences are realized. Denote by $s_{i,k}^*$ the true state for agent $P_i$ at stage $k$. Next, arms display their interests to all agents. For example, students apply to colleges in a given period. Under the assumption that students incur negligible application costs, submitting applications to all colleges is the dominant strategy as students lack information on how colleges evaluate their academic ability or personal essays [6, 13]. Next, at each stage $k \in [K]$, agents simultaneously pull available arms that have not previously rejected them. Each arm either accepts one of the agents that pulled it (if any) or rejects all. An arm exits the market once it accepts an agent, and agents are allowed to exit the market at any time. The arms act simultaneously at each stage. They cannot "hold" offers for accepting or rejecting at a later stage. Hence, agents make "exploding" offers, and arms have to make *irreversible* decisions without knowing what other offers are coming in later stages. Finally, this multi-stage matching process ends when all agents have exited or when a pre-specified number of stages has been reached. If there remain arms in the market when the matching has terminated, these arms are unmatched.

**Agent's expected payoff**    An agent's goal is to maximize the expected payoff, which consists of two parts: the expected utilities and the penalty for exceeding the quota. Let $\mathcal{A}_k$ be the set of arms that are available in the market at stage $k \in [K]$. Suppose that agent $P_i$ pulls arms from the set

$\mathcal{B}_{i,k} \subseteq \{\mathcal{A}_k \setminus \cup_{l \leq k-1} \mathcal{B}_{i,l}\}$ at stage $k$, where $\mathcal{A} \setminus \mathcal{B}$ denotes that set $\mathcal{A}$ minus set $\mathcal{B}$. Let $\mathcal{C}_{i,k} \subseteq \mathcal{B}_{i,k}$ be the set of arms that accept $P_i$ at stage $k$. Then $\mathcal{C}_{i,k}$ is unknown until stage $k+1$, where $k \leq K-1$, and $\mathcal{C}_{i,K}$ is unknown until the end of the matching process. Then $P_i$'s expected payoff at stage $k \in [K]$ is lower bounded by

$$\mathcal{U}_{i,k}[\mathcal{B}_{i,k}] \equiv \sum_{j \in \mathcal{B}_{i,k}} (v_j + e_{ij}) \, \pi_{i,k}(s_{i,k}^*, v_j) - \gamma_i \max\{\mathcal{N}_{i,k}(\mathcal{B}_{i,k}) - q_i, \, 0\}. \tag{2}$$

Here $\mathcal{N}_{i,k}(\mathcal{B}_{i,k}) \equiv \sum_{j \in \mathcal{B}_{i,k}} \pi_{i,k}(s_{i,k}^*, v_j) + \mathrm{card}(\cup_{l \leq k-1} \mathcal{C}_{i,l})$, and $s_{i,k}^*$ is the true state for agent $P_i$ at stage $k$. We assume that the marginal penalty $\gamma_i$ satisfies $\gamma_i > \max_{j \in \mathcal{A}}\{v_j + e_{ij}\}$, which implies that the penalty is greater than arm's latent utility. Since our model involves unknown strategies of the opponents and uncertain arms' preferences, we consider the optimal expected payoff in (2) instead of the optimal realized payoff. Similar expected payoff have been studied in [13, 15].

# 3 Statistical Learning of the Optimal Strategy

We consider a variational formulation of the optimal strategy in Section 3.1 and propose a two-step algorithm using a statistical machine learning method in Section 3.2.

## 3.1 Variational formulation

The problem of finding the optimal set of arms, and the corresponding optimal value $\bar{\mathcal{U}}_i$, can be described as follows:

$$\bar{\mathcal{U}}_i = \max_{\mathcal{B}_{i,k} \subseteq \{\mathcal{A}_k \setminus \cup_{l \leq k-1} \mathcal{B}_{i,l}\}, k \in [K]} \sum_{k \in [K]} \mathcal{U}_{i,k}[\mathcal{B}_{i,k}], \tag{3}$$

where the expected payoff $\mathcal{U}_{i,k}$ is defined in (2). Finding and checking an optimal solution to (3) is difficult. Suppose that an arm set $\cup_{k \in [K]} \bar{\mathcal{B}}_{i,k}$ is given and that it is claimed to be the optimal solution to (3). It is clear that the problem of verifying that $\cup_{k \in [K]} \bar{\mathcal{B}}_{i,k}$ is optimal is computationally intractable; because we need to individually check a significant fraction of the combinations of $\mathrm{card}(\cup_{k \in [K]} \mathcal{A}_k)$ arms to determine which combination might give a larger expected payoff than the given arm set $\cup_{k \in [K]} \bar{\mathcal{B}}_{i,k}$. Since the number of combinations grows exponentially with the number of arms, the complexity of any systematic algorithm becomes impractically large. Moreover, the expected payoff $\mathcal{U}_{i,k}$ depends on the unknown true state $s_{i,k}^*$, which creates yet another layer of difficulty for finding and checking an optimal solution.

**Variational problem** We introduce the following notation: $\delta_{i,k}(v) \equiv \frac{1}{2}[\max_{s_{i,k}} \pi_{i,k}(s_{i,k}, v) - \min_{s_{i,k}} \pi_{i,k}(s_{i,k}, v)]$, which measures the uncertainty of the acceptance probability with respect to the unknown state. Using this notation, we show that a variational formulation gives a practical methodology for finding the optimal strategy.

**Theorem 1.** *There exist parameters $\eta_{i,k} > 0$, for $k \leq K-1$, and $\eta_{i,K} = 0$ such that with high probability, the minimizer of the following variational loss, $\forall k \in [K]$,*

$$\mathcal{L}_{i,k}^{\dagger}[\mathcal{B}_{i,k}] = \sum_{j \in \mathcal{B}_{i,k}} (v_j + e_{ij}) \left[\eta_{i,k}\delta_{i,k}(v_j) - \pi_{i,k}(s_{i,k}^*, v_j)\right] + \gamma_i \max\{\mathcal{N}_{i,k}(\mathcal{B}_{i,k}) - q_i, \, 0\}, \quad (4)$$

*gives a maximizer of the total expected payoff $\sum_{k=1}^{K} \mathcal{U}_{i,k}[\mathcal{B}_{i,k}]$. Here the expected payoff $\mathcal{U}_{i,k}[\mathcal{B}_{i,k}]$ is given in (2), and $\mathcal{B}_{i,k} \subseteq \{\mathcal{A}_k \setminus \cup_{l \leq k-1} \mathcal{B}_{i,l}\}$ for any $k \in [K]$.*

We make four remarks regarding this theorem. First, the parameter $\eta_{i,k} \geq 0$ in (4) is induced by the *hierarchical structure* in the sense that the arms available at subsequent stages are worse than the current ones; see Appendix B.1. Hence, each agent prefers arms with a stable acceptance probability, and for which $\eta_{i,k}$ controls the penalty on the uncertainty. Second, $\eta_{i,k}$ serves as a regularization parameter in the optimization (4) for the uncertainty measure $\delta_{i,k}$. In practice, we may choose a large value of $\eta_{i,k}$ if the agents' competition is tense, as the arms available at subsequent stages are much worse than the current ones. Third, we note that the multi-stage decentralized matching problem is different from the multi-armed bandit problem [10, 25, 26]. A bandit problem is a sequential

allocation problem in which an environment repeatedly provides an agent with a fixed set of arms. Although similar in that it involves sequential decision making under limited information, the multi-stage matching market involves multiple agents competing for arms. An arm exits the market once it accepts an offer. The competition induces a hierarchical structure which makes the optimization in (4) different from the optimization in multi-armed bandits. Finally, there exists a fundamental difference between the multi-stage matching when $K > 1$ and the single-stage matching when $K = 1$. In particular, when $K > 1$, the optimization (4) has a regularization term $\eta_{i,k}\delta_{i,k}(v_j) > 0$ on the uncertainty of the acceptance probability. In contrast, this term vanishes when $K = 1$ as $\eta_{i,K} = 0$. As a result, the optimal strategy in multi-stage matching in Section 3.2 and its economic consequences in Section 4 are distinct from those in single-stage matching [15].

**Greedy strategy**   Although the variational problem in (4) requires only stage-wise optimization and can be solved sequentially for each $k \in [K]$, the finding and checking of an optimal solution is still computationally intractable. This is because we need to individually check a significant fraction of the combinations of $\text{card}(\mathcal{A}_k \setminus \cup_{l \leq k-1}\mathcal{B}_{i,l})$ arms at each stage $k \in [K]$ to determine the optimal solution for (4). The number of combinations grows exponentially with $\text{card}(\mathcal{A}_k \setminus \cup_{l \leq k-1}\mathcal{B}_{i,l})$ for $k \in [K]$.

We propose a greedy algorithm that gives an approximate solution to the optimization problem in (4). Suppose the true state is fixed at $s_{i,k}^* = s_{i,k}$. We refer to $(v_j + e_{ij})[\pi_{i,k}(s_{i,k}, v_j) - \eta_{i,k}\delta_{i,k}(v_j)]$ as arm $A_j$'s *variational expected utility*. For each $A_j \in \{\mathcal{A}_k \setminus \cup_{l \leq k-1}\mathcal{B}_{i,l}\}$, the greedy algorithm computes the variational expected utility per unit of acceptance probability, that is,

$$r(A_j) \equiv (v_j + e_{ij})[\pi_{i,k}(s_{i,k}, v_j) - \eta_{i,k}\delta_{i,k}(v_j)]/\pi_{i,k}(s_{i,k}, v_j).$$

Then the algorithm ranks arms according to its associated value of $r$ so that $r_{(1)} \geq r_{(2)} \cdots \geq r_{(\text{card}(\mathcal{A}_k \setminus \cup_{l \leq k-1}\mathcal{B}_{i,l}))}$. Starting with the first arm corresponding to $r_{(1)}$ and continuing in order, the algorithm selects the arm if its variational expected utility is larger than the expected penalty of exceeding the quota. This algorithm terminates when it arrives at a cutoff value of $r$. Then only arms whose associated $r$ value are better than or equal to the cutoff are selected for agent $P_i$ to pull at stage $k \in [K]$. We present the formalized cutoff $r = r_*$ in Appendix B.2. Then using the greedy algorithm, agent $P_i$ pulls arms from the following set,

$$\widehat{\mathcal{B}}_{i,k}(s_{i,k}) = \left\{ j \mid A_j \in \{\mathcal{A}_k \setminus \cup_{l \leq k-1}\mathcal{B}_{i,l}\} \text{ satisfying } r(A_j) \geq r_* \right\}. \tag{5}$$

**Theorem 2.** *Suppose the true state is fixed at $s_{i,k}^* = s_{i,k}$. The arm set $\widehat{\mathcal{B}}_{i,k}(s_{i,k})$ in (5) is near-optimal as its loss satisfies*

$$\min_{\mathcal{B}_{i,k} \subseteq \{\mathcal{A}_k \setminus \cup_{l \leq k-1}\mathcal{B}_{i,l}\}} \mathcal{L}_i^\dagger[\mathcal{B}_{i,k}] \leq \mathcal{L}_i^\dagger[\widehat{\mathcal{B}}_{i,k}(s_{i,k})] \leq \min_{\mathcal{B}_{i,k} \subseteq \{\mathcal{A}_k \setminus \cup_{l \leq k-1}\mathcal{B}_{i,l}\}} \mathcal{L}_i^\dagger[\mathcal{B}_{i,k}] + UE^\dagger,$$

*where the loss function $\mathcal{L}_i^\dagger$ is defined in (4). The quantity $UE^\dagger \geq 0$ and it equals 0 if there is a continuum of arms and $\pi_{i,k}(\cdot, v)$ is continuous in $v$.*

### 3.2   A two-step learning algorithm

Since the true state and the acceptance probability are unknown a priori in practice, the greedy strategy in (5) is unknown a priori to the agent $P_i$. We propose a two-step algorithm to learning the greedy strategy by using historical data and statistical machine learning methods. The two-step algorithm is built upon the concepts of *lower uncertainty bound* (LUB) and *calibrated decentralized matching* (CDM) [15]. In the first step, we compute an estimated expected utility of each arm and its lower uncertainty bound. Many machine learning methods can be applied here for the modeling of historical data. In the second step, we calibrate the state parameter in a data-driven approach that takes the opportunity cost and penalty for exceeding the quota into account. Based on the calibrated state, an agent selects arms with the largest lower uncertainty bounds of the expected utility. The key idea is to select arms which have large expected utility or little uncertainty in the expected utility.

**Step 1: Lower uncertainty bound**   Let $\mathcal{A}^t = \{A_1^t, A_2^t, \ldots, A_{n^t}^t\}$ be the arm set at $t \in [T] \equiv \{1, \ldots, T\}$. Let $s_{i,k}^t$ be the state of agent $P_i$ at stage $k$ and time $t$. The state $s_{i,k}^t$ is *unknown* until the next stage or the next time point, and the state $s_{i,k}^t$ varies over time. For instance, the yield rate of a college may change over the years. For any arm $A_j^t \in \mathcal{A}^t$, there are an associated pair of the

score and fit values $(v_j^t, e_{ij}^t)$ obtained from (1), where $i \in [m], j \in [n^t]$. Let $(v_j^t, e_{ij}^t)$ denote the attributes of arm $A_j^t$. Define the set $\mathcal{B}_{i,k}^t = \{j \mid P_i \text{ pulls arm } A_j^t \text{ at time } t \text{ and step } k, 1 \leq j \leq n^t\}$, where $\text{card}(\mathcal{B}_{i,k}^t) = n_{i,k}^t \leq n^t$. For any $j \in \mathcal{B}_{i,k}^t$, the outcome that $P_i$ observes is whether an arm $A_j^t$ accepted $P_i$, that is, $y_{ij}^t = \mathbf{1}\{A_j^t \text{ accepts } P_i\}$. We want to estimate $\pi_{i,k}$ based on the historical data, $\mathcal{D} = \{(s_{i,k}^t, v_j^t, e_{ij}^t, y_{ij}^t) : i \in [m]; j \in \cup_{k=1}^K \mathcal{B}_{i,k}^t; t \in [T]\}$.

A wide range of machine learning methods, e.g., reproducing kernel methods, random forests, or neural networks, can be applied here to learn $\pi_{i,k}$ (cf. [23]). For concreteness, we consider a penalized estimator in RKHS. Let the log odds ratio $f_{i,k}(s_{i,k}, v) = \log\{\pi_{i,k}(s_{i,k}, v)/[1 - \pi_{i,k}(s_{i,k}, v)]\}$, which is assumed to reside in an RKHS $\mathcal{H}_{\mathcal{K}_{i,k}}$ with the kernel $\mathcal{K}_{i,k}$. Then we solve for $\widehat{f}_{i,k} \in \mathcal{H}_{\mathcal{K}_{i,k}}$ that minimizes the objective function:

$$\sum_{t=1}^T \frac{1}{n_{i,k}^t} \sum_{j \in \mathcal{B}_{i,k}^t} \left[-y_{ij}^t f_{i,k}(s_{i,k}^t, v_j^t) + \log\left(1 + \exp\left(f_{i,k}(s_{i,k}^t, v_j^t)\right)\right)\right] + \lambda_{i,k}\|f_{i,k}\|_{\mathcal{H}_{\mathcal{K}_{i,k}}}^2,$$

where $\lambda_{i,k} \geq 0$ is a tuning parameter. Consider the tensor product structure of $\mathcal{H}_{\mathcal{K}_{i,k}}$, where $\mathcal{K}_{i,k}((s_i, v), (s_i', v')) = \mathcal{K}_{i,k}^s(s_i, s_i')\mathcal{K}_{i,k}^v(v, v')$ with some kernel functions $\mathcal{K}_{i,k}^s$ and $\mathcal{K}_{i,k}^v$ [40]. It is known that $\widehat{f}_{i,k}$ is minimax rate-optimal and satisfies $\mathbb{E}[(\widehat{f}_{i,k} - f_{i,k})^2] \leq c_f[T(\log T)^{-1}]^{-2r/(2r+1)}$ for any $i \in [m]$ (cf. [15]). Here, $c_f > 0$ is a constant independent of $T$, and $r \geq 1$ denotes the order of smoothness. The value of learning from historical data is particularly significant when a new arm is introduced into the problem. Let $\mathcal{A}^{T+1} = \{A_1, \ldots, A_n\}$ be the new arm set at time $T + 1$, where $A_j$ has attributes obtained from (1). Then the probability that $A_j$ accepts $P_i$ at stage $k$ is estimated by $\widehat{\pi}_{i,k}(s_{i,k}, v_j) = \{1 + \exp[-\widehat{f}_{i,k}(s_{i,k}, v_j)]\}^{-1}$. The expected utility of $A_j$ is $\widehat{\pi}_{i,k}(s_{i,k}, v_j)(v_j + e_{ij})$ for any $j \in [n]$. Finally, we construct a lower uncertainty bound for $\pi_{i,k}(s_{i,k}, v_j)$ as,

$$\widehat{\pi}_{i,k}^{\mathrm{L}}(s_{i,k}, v_j) = \begin{cases} \widehat{\pi}_{i,k}(s_{i,k}, v_j) - \eta_{i,k}\widehat{\delta}_{i,k}(v_j), \\ \quad \text{if } v_j \in [\min\{v_j^t \mid j \in \cup_{t=1}^T \mathcal{B}_i^t\}, \max\{v_j^t \mid j \in \cup_{t=1}^T \mathcal{B}_i^t\}]; \\ 1, \quad \text{o.w.,} \end{cases} \tag{6}$$

where $\widehat{\delta}_{i,k}(v_j) = \frac{1}{2}[\max_{s_{i,k}} \widehat{\pi}_{i,k}(s_{i,k}, v_j) - \min_{s_{i,k}} \widehat{\pi}_{i,k}(s_{i,k}, v_j)]$. The parameter $\eta_{i,k} \geq 0$ is defined in (4). Note that (6) assigns probability one to arms with scores that agent $P_i$ has never pulled. Hence it encourages the exploration of previously untried arms. A *lower uncertainty bound* for the expected utility is then given by $\widehat{\pi}_{i,k}^{\mathrm{L}}(s_{i,k}, v_j)(v_j + e_{ij})$ for any $j \in [n]$.

The prediction of match compatibility is also possible in another direction that an arm $A_j$ can also learn how much an agent $P_i$ may like itself by predicting the probability that $A_j$ can be pulled by $P_i$. The arms would make the decisions based on the prediction that if they have a realistic potential of being pulled by a better agent. This feature also distinguishes the two-sided matching platform from a one-sided recommendation engine that only considers which arms an agent may like, but not which arms may also like the agent in return.

**Step 2: Calibrated decentralized matching** Since the true state $s_{i,k}^*$ is unknown in practice, a natural question is how to calibrate the state parameter $s_{i,k}$ in (5). Consider the average-case loss, $\mathbb{E}_{s_{i,k}^*}\{\mathcal{L}_{i,k}^\dagger[\widehat{\mathcal{B}}_{i,k}(s_{i,k})]\}$, where the loss $\mathcal{L}_{i,k}^\dagger$ is defined in (4). Define the *marginal set* as $\partial\widehat{\mathcal{B}}_{i,k}(s_{i,k}) \equiv \lim_{\delta_s \to 0_+}\{\widehat{\mathcal{B}}_{i,k}(s_{i,k} - \delta_s) \setminus \widehat{\mathcal{B}}_{i,k}(s_{i,k})\}$. Hence $\partial\widehat{\mathcal{B}}_{i,k}(s_{i,k})$ represents the change of $\widehat{\mathcal{B}}_{i,k}(s_{i,k})$ with a perturbation of $s_{i,k}$.

**Theorem 3.** *The average-case loss* $\mathbb{E}_{s_{i,k}^*}\{\mathcal{L}_{i,k}^\dagger[\widehat{\mathcal{B}}_{i,k}(s_{i,k})]\}$ *is minimized if* $s_{i,k} \in (0,1)$ *is chosen as the solution to*

$$\mathbb{P}(s_{i,k}^* \neq s_{i,k}) \sum_{j \in \partial\widehat{\mathcal{B}}_{i,k}(s_{i,k})} (v_j + e_{ij})\mathbb{E}_{s_{i,k}^*}\left[\pi_{i,k}(s_{i,k}^*, v_j) - \eta_{i,k}\delta_{i,k}(v_j) \mid s_{i,k}^* \neq s_{i,k}\right]$$
$$= \gamma_i[1 - F_{s_{i,k}^*}(s_{i,k})] \sum_{j \in \partial\widehat{\mathcal{B}}_{i,k}(s_{i,k})} \mathbb{E}_{s_{i,k}^*}[\pi_{i,k}(s_{i,k}^*, v_j) \mid s_{i,k} < s_{i,k}^* \leq 1], \tag{7}$$

*where* $F_{s_{i,k}^*}$ *is the cumulative distribution function of* $s_{i,k}^* \in [0,1]$.

---

**Algorithm 1** The two-step algorithm for multi-stage decentralized matching

---

1: **Inputs**: Historical data for an agent $P_i$: $\{(s_{i,k}^t, v_j^t, e_{ij}^t, y_{ij}^t) : j \in \mathcal{B}_{i,k}^t; t = 1, 2, \ldots, T\}$; New
   arm set $\mathcal{A}^{T+1}$ at time $T + 1$, where the arms have attributes $\{(v_j, e_{ij}) : j \in [n]\}$; Penalty $\gamma_i$ for
   exceeding the quota. Regularization parameter $\eta_{i,k} \geq 0$.
2: **for** stage $k = 1, 2, \ldots, K$ **do**
3:    Construct the lower uncertainty bound $\widehat{\pi}_{i,k}^{\mathrm{L}}(s_{i,k}, v_j)$ by (6)
4:    Estimate the distribution $F_{s_{i,k}^*}(\cdot)$ by the kernel density method [38]
5:    Calibrate the state $s_{i,k}$ according to Theorem 3
6:    Determine the arm set $\widehat{\mathcal{B}}_{i,k}^{\mathrm{L}}(s_{i,k})$ in (8)
7:    Calculate the remaining quota: $q_i - \mathrm{card}(\cup_{l \leq k-1} \mathcal{C}_{i,l})$ and the available arms
8: **end for**
9: **Outputs**: The arm set $\widehat{\mathcal{B}}_{i,k}^{\mathrm{L}}(s_{i,k})$ for agent $P_i$ at each stages.

---

The key idea of (7) is to balance the trade-off between opportunity cost and penalty for exceeding the quota. If (7) has more than one solution, then $s_{i,k}$ is chosen as the largest one. If the distribution $F_{s_{i,k}^*}$ has discrete support, the objective in Theorem 3 needs to be changed as follows: choosing the minimal $s_{i,k} \in [0, 1]$ such that the left side of (7) is not less than the right side of (7), where the search of $s_{i,k}$ starts from the maximum value in the support and decreases to the minimal value. Moreover, instead of the average-case loss in Theorem 3, we can also perform the calibration under the worst-case loss, which is discussed in Appendix B.3.

**Summary of the two-step algorithm**   Using (6) and (7), we can obtain the cutoff estimate $\widehat{r}_*$ and calibrated state $s_{i,k}$, which suggests agent $P_i$ to pull arms from the following set at stage $k$:

$$\widehat{\mathcal{B}}_{i,k}^{\mathrm{L}}(s_{i,k}) = \left\{ j \mid A_j \in \{\mathcal{A}_k^{T+1} \setminus \cup_{l \leq k-1} \mathcal{B}_{i,l}\} \text{ satisfying } r(A_j) \geq \widehat{r}_* \right\}. \tag{8}$$

Here $\mathcal{A}_k^{T+1}$ is the set of arms that are available at stage $k$ of time $T + 1$. Due to the minimax optimality of $\widehat{f}_{i,k}$, we have the consistency result that $\widehat{\mathcal{B}}_{i,k}^{\mathrm{L}}(s_{i,k}) \to \widehat{\mathcal{B}}_{i,k}(s_{i,k})$ as $T \to \infty$, where the set $\widehat{\mathcal{B}}_{i,k}(s_{i,k})$ is defined in (5). We summarize the above two-step algorithm in Algorithm 1. We also remark that although the negligible application costs is assumed in Section 2, Algorithm 1 is applicable to non-negligible application costs, in which different agents (i.e., colleges) would have different sets of available arms (i.e., student applicants).

## 4   Strategic Behavior and Economic Implications

Agents in a multi-stage decentralized matching markets cannot observe other agents' quotas or the choices of the arms that accept other agents. Each agent only observes the arms that are left in the market at each stage. Theorem 1 implies that agents prefer arms with stable acceptance probability. This preference lead to strategic behavior on the part of the agents as follows. Define the *uncertainty level* as the uncertainty measure $\delta_{i,k}(v)$ in Section 3.1 relative to the acceptance probability $\pi_{i,k}(s_{i,k}, v)$. That is,

$$\text{uncertainty level} \equiv \delta_{i,k}(v) / \pi_{i,k}(s_{i,k}, v). \tag{9}$$

We show in Appendix B.2 that the cutoff $r_*$ in (5) is strictly increasing in the uncertainty level for any $v \in [0, 1]$ and $k \leq K - 1$, which implies that an agent favors arms with a low uncertainty level. Hence, an agent's *strategic behavior* in this market is to strategically select arms with a low uncertainty level. We now study the implications of such strategic behavior on fairness and welfare.

**No justified envy**   The fairness studied here is defined in terms of *no justified envy* [1, 8]. Specifically, an arm $A_j$ has justified envy if, at a stage $k \in [K]$, $A_j$ prefers an agent $P_{i'}$ to another agent $P_i$ that pulls $A_j$, even though $P_{i'}$ pulls an arm $A_{j'}$ which ranks below $A_j$ according to the true preference of $P_{i'}$. We define a multi-stage matching procedure to be *fair* if there is no arm having justified envy at any stage.

**Proposition 1.** *The probability that an arm has justified envy is strictly increasing in the arm's uncertainty level defined in (9).*

The fairness issue has been noted in practical multi-stage matching markets. For example, candidates in job markets may "fall through the cracks"—an employer that values a candidate highly perceives that the candidate is unlikely to accept the job offer and hence declines to conduct an interview with the candidate; hence, candidates may have justified envy [14]. Besides our ex-ante definition of no justified envy, there are other choices of no justified envy, including ex-post definition, which could lead to a different set of technical results [19].

**Fairness vs. welfare trade-off**   We note that by Theorem 1, an agent has increased expected payoff under $\eta_{i,k} > 0$ than under $\eta_{i,k} = 0$ for all stages $k \leq K - 1$. Define the number of arms with justified envy to be the *level of justified envy* of the matching outcome. Then if the level of justified envy is zero, the matching outcome is fair for arms.

**Proposition 2.** *The level of justified envy is strictly increasing in $\eta_{i,k} \geq 0$.*

This proposition implies a trade-off between welfare and fairness since both the level of justified envy and welfare increase when changing $\eta_{i,k} = 0$ to $\eta_{i,k} > 0$. We give an example of two-stage decentralized matching, that is, $K = 2$. Such two-stage matching is typical in college admissions, which may include regular admissions and waiting-list admissions. By Theorems 1 and 2, agents in the first stage would strategically pull arms with low uncertainty levels by taking $\eta_{i,1} > 0$. In this way, agents would reduce head-on competition. Next, agents in the second stage would act according to their true preferences and pull available arms with top latent utilities by taking $\eta_{i,2} = 0$. Theorem 1 shows that agents' strategic behavior in the first stage increases the welfare compared to acting according to their true preferences, whereas in the second stage, agents acting according to their true preferences suffices. Proposition 2 shows that agents' strategic behavior in the first stage results in increased welfare, but at the cost of arms' fairness.

**Comparison with single-stage matching markets**   Different from multi-stage matching markets, the optimal strategy in single-stage matching gives a fair outcome for arms [15]. However, we show that agents are better off in multi-stage markets compared to single-stage markets.

**Proposition 3.** *Agents have improved welfare under multi-stage decentralized matching than under single-stage decentralized matching.*

We provide an empirical example in Appendix A.4 to illustrate the gap between multi-stage welfare and single-stage welfare.

**Comparison with centralized matching markets**   Many centralized matching markets are implemented by employing the celebrated deferred acceptance (DA) algorithm [21]; see examples in [1, 32]. In the arm-proposing version of DA (e.g., student-proposing in college admissions), agents and arms report their ordinal preferences to a clearinghouse, which simulates the following multi-stage procedure. Every arm shows its interest to the most preferred agent that has not yet rejected it at each stage. Every agent tentatively pulls the most preferred arms up to its quota limit and permanently rejects the remaining arms that have indicated their interest to the agent. Once the process terminates, each arm is assigned to the agent that has tentatively pulled it or otherwise remains unmatched. The multi-stage decentralized matching is different from DA in practice, mainly due to the acceptance is not tentative (i.e., non-deferrable) in decentralized matching. Moreover, there is usually a restriction on the number of stages in decentralized matching due to the time cost at each stage of multi-stage decentralized matching is not negligible. We show in a numerical example of Appendix A.3 that some agents are better off in decentralized markets than centralized markets. This finding gives a partial explanation of the prevalence of decentralized college admissions in many countries.

## 5   Numerical Studies

In this section we demonstrate aspects of the theoretical predictions through a simulation and a real data application in college admissions. We provide extensive numerical comparisons of Algorithm 1 with other methods in Appendix. We also give additional real data analysis in Appendix. The total computing hour is within one hour in personal laptop with Intel Core i5.

**Simulated graduate school admissions**  Consider 50 graduate schools from three tiers of colleges: five top colleges $\{P_1, \ldots, P_5\}$, ten good colleges $\{P_6, \ldots, P_{15}\}$, and 35 other colleges $\{P_{16}, \ldots, P_{50}\}$. Each has the same quota $q = 5$ and penalty $\gamma = 2.5$. The simulation generates students' preferences with ten different states $\{s_1, \ldots, s_{10}\} \subset [0, 1]$. For any state, students' preferences for colleges from the same tier are random. However, students prefer top colleges to good colleges, and the other colleges are the least favorite. The random preferences depend on the state due to colleges' uncertain reputation and popularity in the current year. We consider varying numbers of students $\{250, 260, 270, 280, 290, 300\}$. For each size of students, there are ten students having score $v_j$ chosen uniformly and i.i.d. from $[0.9, 1]$ and 100 students having score $v_j$ i.i.d. uniformly chosen from $[0.7, 0.9]$. The rest of the students have score $v_j$ randomly chosen from $[0, 0.7]$. The fits $e_{ij}$ for all college-student pairs are drawn uniformly and i.i.d. from $[0, 1]$.

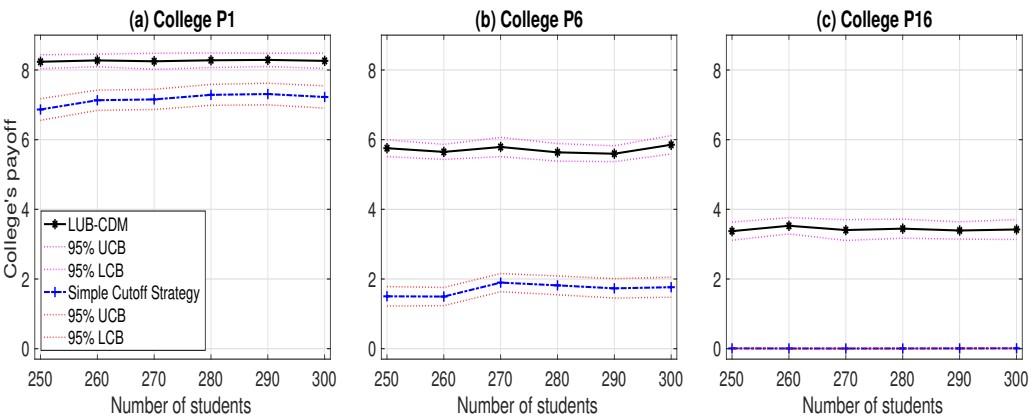

Figure 1: Performance of the proposed Algorithm 1 (i.e., LUB-CDM) and the Simple Cutoff Strategies with varying numbers of students. The results are averaged over 500 data replications. (a): College $P_1$ from tier 1. (b): College $P_6$ from tier 2. (c): College $P_{16}$ from tier 3.

We compare the college's expected payoff achieved by the proposed Algorithm 1 with the *simple cutoff strategy*, where the latter method has each college choosing the most preferred students up to the remaining quota at each stage. The training data are simulated from colleges' random proposing by pulling a random number of arms according to the latent utilities. The training data consists of 20 times of random proposing under each of the arms' preference structures with the two-stage admissions. This training data simulates the graduate school admissions over 20 years. The testing data draws a random state from $\{s_1, \ldots, s_{10}\}$ which gives the corresponding arms' preferences. Then we apply Algorithm 1 with $\eta_{i,1} = 0.1, \eta_{i,2} = 0$ and $\gamma_i = 2.5$. Figure 1 reports the averaged payoffs of three colleges $P_1, P_6,$ and $P_{16}$ over 500 data replications. Here colleges $P_1, P_6,$ and $P_{16}$ belong to the three different tiers, respectively. In Figure 1, all colleges except $P_1$ use Algorithm 1 while $P_1$ uses one of the two methods: Algorithm 1 and the simple cutoff strategy. It is seen that Algorithm 1 gives the largest average payoffs for all of $P_1, P_6$ and $P_{16}$. In particular, Algorithm 1 performs significantly better for $P_6$ and $P_{16}$ compared to the simple cutoff strategy.

**U.S. college admissions**  We study a public data on college admissions from the *New York Times* "The Choice" blog. In this dataset, 37 U.S. colleges reported their admission yields and waiting list offers for 2015–17 applicants without personally identifiable information. As we discussed in Section 2, a college's yield is a proxy for the state $s_{i,k}$ as it indicates the college's popularity. The set of 37 colleges consists of liberal arts colleges, national universities, and other undergraduate programs.

We estimate the uncertainty level $\delta_{i,k}(v)\pi_{i,k}^{-1}(s_{i,k}, v)$ defined in (9) and study colleges' strategic responses. While conclusive evidence on the individual students' acceptance probability is difficult to obtain, we estimate the college-wise uncertainty on the yield: $\sqrt{\text{Var}(s_{i,k})}s_{i,k}^{-1}$. Since the choice set for admitted students differs across years, the yield's uncertainty underestimates the uncertainty facing a college. Figure 2 shows that colleges' uncertainty levels are much smaller than one, which, together with Theorem 1, implies that students face limited unfairness. In particular, the yield uncertainty is robust to the size of admitted students; see the left plot of Figure 2. On the other hand, top-ranked national universities may have higher uncertainty levels; see the right plot of Figure 2, where the

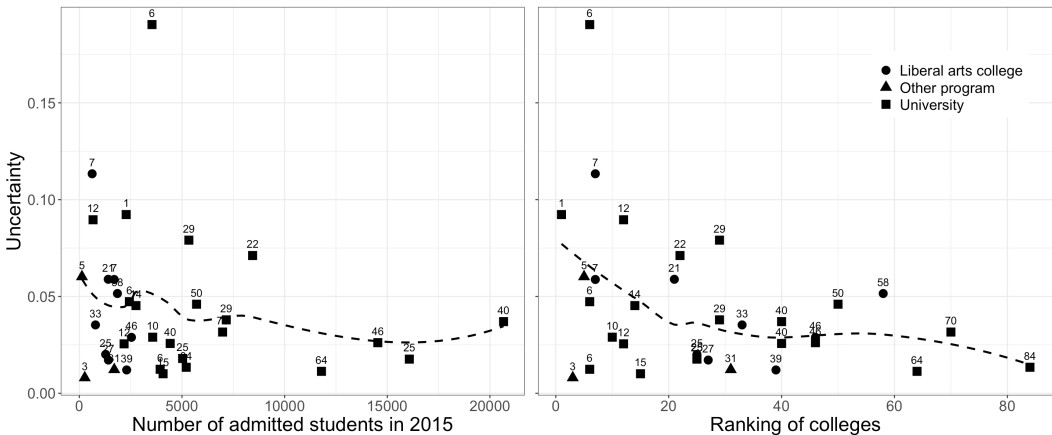

Figure 2: Regression of uncertainty level on the size of admitted class and the ranking, respectively. Two dashed curves are fitted using smoothing splines with the tuning parameter chosen by GCV.

outlier is the University of Chicago at the .19 uncertainty level. We verify the higher uncertainty level for top universities using the waiting list data. We perform Fisher's exact test for the rank data on the difference of rates of accepted waiting list students to total enrolled students over 2015–16. This statistic reflects the uncertainty on both the regular admission yield and the wait-listed students' quality. We reject the null hypothesis that the uncertainty of acceptance is the same for all national universities at the .05 significance level. The higher uncertainty for top-ranked national universities may arise due to the intense competition. Those universities are better off by employing strategic admission to reduce the enrollment uncertainty. This result implies that students are more likely to experience unfairness when applying for top national universities.

# 6 Conclusion

This paper develops a nonparametric statistical model to learn optimal strategies in multi-stage decentralized matching markets. The model provides insight into the interplay between learning and economic objectives in decentralized matching markets. In the model, arms have uncertain preferences that depend on the unknown state of the world and competition among the agents. We propose an algorithm, built upon the concepts of lower uncertainty bound and calibrated decentralized matching, for learning optimal strategies using historical data. We find that agents act strategically in favor of arms with low uncertainty levels of acceptance. The strategic targeting improves an agent's welfare but leads to unfairness for arms. Our theory allows analytical comparisons between single-stage decentralized markets and centralized markets.

For future directions, it is of interest to study algorithmic strategies when agents' preferences show complementarities or indifference. These settings have important applications, as firms may demand workers that complement one another in terms of their skills and roles, or some applicants are indistinguishable to a firm. We leave these questions for future work.

The problem of machine learning in economics has become increasingly important in many application domains. In this work, we aim to deepen the understanding of decentralized matching markets from a learning perspective and propose an efficient and scalable algorithm to solve optimal strategies. We do not foresee any negative impact to society from our work.

## Acknowledgments and Disclosure of Funding

We would like to thank the area chair and four anonymous referees for constructive suggestions that improve the paper. We thank Robert M. Anderson and Joel Sobel for helpful discussions. This work was supported in part by the Vannevar Bush Faculty Fellowship program under grant number N00014-21-1-2941.

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
