# A Supplementary Numerical Results

## A.1 Comparison with the straightforward strategy

In this example, we compare the proposed Algorithm 1 with the *straightforward strategy*, where the latter method pulls arms according to the latent utility defined in Eq. (1) and calibrates the state in the same way as Algorithm 1.

Suppose there are $n$ arms $\mathcal{A} = \{A_1, A_2, \ldots, A_n\}$ and three agents $\mathcal{P} = \{P_1, P_2, P_3\}$, where each agent has a quota $q < n/3$. There are two equally likely states: $s_a$ and $s_b$ with $s_a = 1 - s_b > 1/2$. All arms prefer $P_1$ and $P_2$ to $P_3$, but the arms prefer $P_3$ compared to being unmatched. Agents $P_1$ and $P_2$ evaluate each arm based on score $v$ and with probability $p^* \in (0, 1)$, each of $P_1$ and $P_2$ finds an arm unacceptable. Agent $P_3$ evaluates each arm only based on the score. For each state $j \in \{a, b\}$, a fraction $s_j$ of arms receives utility $u_1$ when matched to $P_1$ and utility $u_2$ when matched to $P_2$, where $u_1 > u_2$ and the remaining $(1 - s_j)$ of arms receive the opposite utilities. Hence, $P_1$ is more popular under the state $s_a$ and $P_2$ is more popular under the state $s_b$. In each state, an arm gets utility $u_3$ from $P_3$, where $(1 - p^*)u_1 < u_3 < u_1$. This condition implies that an arm is better off by accepting $P_3$ than waiting for $P_1$ or $P_2$. We consider a two-stage matching, where at the first stage, each agent pulls a set of arms and wait-lists other arms. An arm pulled by an agent must accept or reject the agent immediately.

**Proposition A.4.** *Agent $P_1$ is better off by using Algorithm 1 than using the straightforward strategy, where the expected payoff is improved by $O(\eta_{1,1})$. Here $\eta_{1,1}$ is the regularization parameter defined in Theorem 1.*

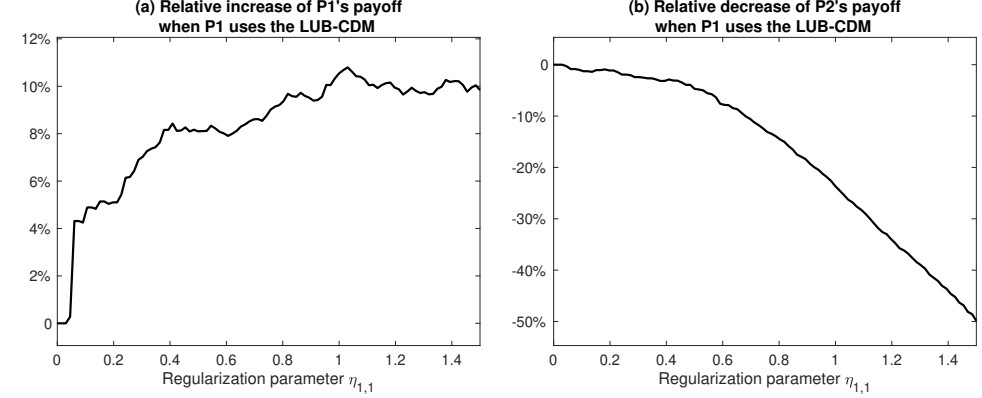

Figure 3: Comparison of the proposed Algorithm 1 (i.e., LUB-CDM) and the straightforward strategy. The results are averaged over 500 data replications. (a) The relative increase of $P_1$'s payoffs when $P_1$ changes from the straightforward strategy to the LUB-CDM, where the improvement is $O(\eta_{1,1})$. (b) The relative decrease of $P_2$'s payoffs when $P_1$ changes from the straightforward strategy to the LUB-CDM.

To illustrate the improvement, we consider the states $s_a = 0.6$, $s_b = 0.4$, the number of arms $n = 100$, the quota $q = 10$, the utilities $u_1 = 1, u_2 = 0.9, u_3 = 0.8$, and the probability $p^* = 0.3$. Suppose that the score $v$ follows a deterministic uniform design points $\{1.05, 1.1, 1.15, \ldots, 2.95, 3\} \subset [1, 3]$. The penalties of exceeding the quota are $\gamma_1 = \gamma_2 = \gamma_3 = 5$. We compare the proposed Algorithm 1 (i.e., LUB-CDM) with the straightforward strategy (i.e., CDM). The latter method is a straightforward strategy as it pulls arms according to the latent utilities in Eq. (1) without strategic behaviors. Figure 3 reports $P_1$'s and $P_2$'s relative changes in payoffs, when $P_1$ changes from using the CDM to using the LUB-CDM. The results are averaged over 500 data replications. Here $P_1$ using the LUB-CDM and the CDM correspond to $\eta_{1,1} > 0$ and $\eta_{1,1} = 0$, respectively. The $P_2$ uses CDM. It is seen the LUB-CDM improves $P_1$'s expected payoff, where the improvement is at the cost of $P_2$'s payoff.

## A.2 Comparison with the patient strategy

In this example, we compare the proposed Algorithm 1 with the *patient strategy*, where the latter method pulls arms according to the latent utility at the beginning stage but has more strategic behaviors as the matching proceeds. We consider a search model due to [2], which captures the search process in matching markets and builds a connection between the multi-stage decentralized matching markets and the centralized matching markets.

Suppose there are $n$ arms $\mathcal{A} = \{A_1, A_2, \ldots, A_n\}$ and $m$ agents $\mathcal{P} = \{P_1, P_2, \ldots, P_m\}$, where each agent has quota $q = 1$. At each stage, each agent comes across a randomly sampled arm. Let $v_{\mathcal{P}}(i)$ and $v_{\mathcal{A}}(j)$ be the reservation utilities of agent $P_i$ and arm $A_j$ from staying unmatched and continuing the search. Recall the latent utility $U_i(A_j)$ in Section 2. Similarly, we define $U_j(P_i)$ as the utility that arm $A_j$ receives when matched to $P_i$. Let $v_{\mathcal{P}}(i)$ and $v_{\mathcal{A}}(j)$ be the reservation utilities of agent $P_i$ and arm $A_j$ from staying single and continuing the search for a match. Hence $\mathbf{1}\{P_i \text{ pulls } A_j\} = \mathbf{1}\{U_i(A_j) \geq v_{\mathcal{P}}(i)\}$, and $\mathbf{1}\{A_j \text{ accepts } P_i\} = \mathbf{1}\{U_j(P_i) \geq v_{\mathcal{A}}(j)\}$. The utility that agent $P_i$ gets upon coming across arm $A_j$ is

$$\bar{U}_i(A_j) = U_i(A_j)\mathbf{1}\{U_i(A_j) \geq v_{\mathcal{P}}(i)\}\mathbf{1}\{U_j(P_i) \geq v_{\mathcal{A}}(j)\}$$
$$+ v_{\mathcal{P}}(i)[1 - \mathbf{1}\{U_i(A_j) \geq v_{\mathcal{P}}(i)\}\mathbf{1}\{U_j(P_i) \geq v_{\mathcal{A}}(j)\}],$$

where the first term on the right-hand side is the utility from a successful match and the second term on the right-hand side is the utility when no match occurs. Adachi's model involves a stage discount factor $\rho > 0$, where the Bellman equations for the optimal reservation values and search rules are

$$v_{\mathcal{P}}(i) = \rho \int \bar{U}_i(A_j)dF_{\mathcal{A}}(j) \quad \text{and} \quad v_{\mathcal{A}}(j) = \rho \int \bar{U}_j(P_i)dF_{\mathcal{P}}(i), \tag{A.10}$$

where $F_{\mathcal{A}}$ and $F_{\mathcal{P}}$ are the distributions that each agent and arm came across. In [2] the author shows that Bellman equations in Eq. (A.10) defines an iterative mapping that converges to the equilibrium reservation utilities $(v_{\mathcal{P}}^*(i), v_{\mathcal{A}}^*(j))$. Furthermore, as $\rho \to 1$, the Bellman equations lead to the matching outcomes that are stable in the sense of Gale and Shapley [21].

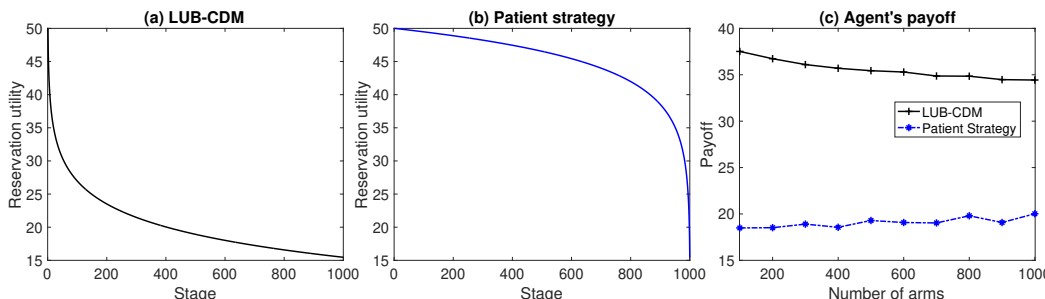

Figure 4: Performance of the proposed Algorithm 1 (i.e., LUB-CDM) and the patient strategy. The results averaged over 500 data replications. (a) $P_1$'s reservation utility under the LUB-CDM given by $50 - 5\log(k)$, where the stage $k = 1, \ldots, 500$. (b) $P_1$'s reservation utility under the patient strategy given by $50 + 5\log(\frac{N+1-k}{N})$. (c) $P_1$'s payoffs with varying number of arms.

Since the equilibrium reservation utilities $(v_{\mathcal{P}}^*(i), v_{\mathcal{A}}^*(j))$ are unknown in practice, agents need to learn an optimal strategy of choosing the reservation utility $v_{\mathcal{P}}(i)$ at different stages. We compare the proposed Algorithm 1 (i.e., LUB-CDM) with the patient strategy, where the latter is defined as the strategy with $\rho = 1$ at the beginning stage $k = 1$ and decreasing $\rho$ as the matching proceeds in Eq. (A.10). Note that LUB-CDM has less strategic behaviors as the matching proceeds. Hence it corresponds to the case that $v_{\mathcal{P}}(i)$ is a convex function of the stages. On the other hand, the patient strategy has more strategic behaviors as the matching proceeds. Hence it corresponds to the case that $v_{\mathcal{P}}(i)$ is a concave function of the stages. Suppose that different arms receive the same utility for matching the same agent, that is, $U_j(P_i) = U_{j'}(P_i), \forall j \neq j'$, which utility is unknown to $P_i$. Similarly, different agents receive the same utility for matching the same arm, that is, $U_i(A_j) = U_{i'}(A_j), \forall i \neq i'$, which utility is known to $A_j$. Then $P_i$ matches with $A_j$ if the event $\{U_j(P_i) \geq U_i(A_j) \geq v_{\mathcal{P}}(i)\}$ holds. Suppose that agent $P_1$'s utility is $U_j(P_1) = 40$, and $m = n \in \{100, 200, \ldots, 1000\}$. Let the reservation utility $v_{\mathcal{P}}(1)$ at the stage $k$ be $50 - 5\log(k)$ and

$50 + 5\log(\frac{N+1-k}{N})$ for the LUB-CDM and the patient strategy, respectively; see Figure 4(a) and (b). Figure 4(c) reports $P_1$'s payoff under two methods, where the LUB-CDM outperforms the patient strategy. Therefore, the strategic behavior at early stages improves the agent's payoff in practice, which result corroborates Theorem 1.

### A.3  Comparison of multi-stage matching and DA

In this example, we compare the multi-stage decentralized matching with the DA algorithm [21]. Suppose there are four arms $\mathcal{A} = \{A_1, A_2, A_3, A_4\}$ and three agents $\mathcal{P} = \{P_1, P_2, P_3\}$. Agents have varied quotas: $q_1 = 2$ and $q_2 = q_3 = 1$. Arms' attributes are given by $v_1 = v_2 = v_3 = 2, v_4 = 1$, and $e_{13} = e_{23} = e_{32} = 0, e_{12} = e_{22} = e_{31} = 0.5, e_{11} = e_{21} = e_{33} = 1, e_{14} = 0.2, e_{24} = 0.5, e_{34} = 0.8$. The latent utilities and arms' true preferences are shown in Table 1. For the decentralized matching, suppose that at each stage, every agent uses the straightforward strategy by pulling its most preferred arms up to the quota. Arms accept their most preferred agent (if any) or wait until the next stage. Then the decentralized matching has the outcome $(A_1, P_1), (A_2, P_1), (A_3, P_3), (A_4, P_2)$. On the other hand, the DA algorithm gives the outcome $(A_1, P_3), (A_2, P_2), (A_3, P_1), (A_4, P_1)$, which the unique stable matching outcome. Here both $P_1$ and $P_3$ strictly prefer the decentralized matching outcome to DA outcome. This result corroborates the remark in Section 4 that some agents are better off under the decentralized matching.

Table 1: (a) Arm's latent utilities for each agent, which corresponds to Eq. (1). (b) Arms' preferences with the number indicating the arms' ranking of agents. For example, $A_1$ ranks $P_3$ first, $P_1$ second, $P_2$ third. These preferences are unknown to agents.

<table>
<tr><th colspan="5">(a) Arm's latent utility</th><th colspan="5">(b) Arm's preference</th></tr>
<tr><th></th><th>$A_1$</th><th>$A_2$</th><th>$A_3$</th><th>$A_4$</th><th></th><th>$A_1$</th><th>$A_2$</th><th>$A_3$</th><th>$A_4$</th></tr>
<tr><td>$P_1$</td><td>3</td><td>2.5</td><td>2</td><td>1.2</td><td>$P_1$</td><td>2</td><td>2</td><td>1</td><td>1</td></tr>
<tr><td>$P_2$</td><td>3</td><td>2.5</td><td>2</td><td>1.5</td><td>$P_2$</td><td>3</td><td>1</td><td>3</td><td>2</td></tr>
<tr><td>$P_3$</td><td>2.5</td><td>2</td><td>3</td><td>1.8</td><td>$P_3$</td><td>1</td><td>3</td><td>2</td><td>3</td></tr>
</table>

Second, we study the incentive of agents in the multi-stage decentralized matching. We show that it is not a dominant strategy for each agent to use the straightforward strategy by pulling arms according to the latent utility. For example, consider the preferences in Table 1. If $P_2$ skips over $A_1$ and firstly pulls $A_2$, and other agents pull their most preferred arms up to their quotas. Then the decentralized matching has the outcome $(A_1, P_1), (A_2, P_2), (A_3, P_3), (A_4, P_1)$, where $P_2$ is strictly better off compared to the outcome when $P_2$ firstly pulls $A_1$.

Table 2: (a) Arm's latent utilities for each agent. (b) Arms' preferences with the number indicating the arms' ranking of agents. For example, $A_1$ ranks $P_4$ first, $P_1$ second, $P_3$ third, $P_2$ fourth.

<table>
<tr><th colspan="5">(a) Arm's latent utility</th><th colspan="5">(b) Arm's preference</th></tr>
<tr><th></th><th>$A_1$</th><th>$A_2$</th><th>$A_3$</th><th>$A_4$</th><th></th><th>$P_1$</th><th>$P_2$</th><th>$P_3$</th><th>$P_4$</th></tr>
<tr><td>$P_1$</td><td>3</td><td>2</td><td>2.6</td><td>2.3</td><td>$A_1$</td><td>2</td><td>4</td><td>3</td><td>1</td></tr>
<tr><td>$P_2$</td><td>2</td><td>2.6</td><td>3</td><td>2.3</td><td>$A_2$</td><td>4</td><td>2</td><td>1</td><td>3</td></tr>
<tr><td>$P_3$</td><td>2.3</td><td>2</td><td>3</td><td>2.6</td><td>$A_3$</td><td>1</td><td>3</td><td>4</td><td>2</td></tr>
<tr><td>$P_3$</td><td>2</td><td>2.3</td><td>2.6</td><td>3</td><td>$A_3$</td><td>3</td><td>1</td><td>2</td><td>4</td></tr>
</table>

Finally, we show that arms can also be better off if they are strategic in multi-stage decentralized matching. Suppose there are four agents and four arms, and each agent has a quota one. The latent utilities and arms' true preferences are given in Table 2. When agents and arms are not strategic, the decentralized matching has the outcome $(A_1, P_1), (A_2, P_3), (A_3, P_2), (A_4, P_4)$. However, suppose arms are strategic, where $A_4$ rejects $P_4$ as $P_4$ is $A_4$'s least favorite agent and $A_4$ believes the coming agent will not be worse. The outcome becomes $(A_1, P_1), (A_2, P_4), (A_3, P_2), (A_4, P_3)$. Hence $A_4$ is strictly better off. Besides, if $A_3$ also rejects $\{P_2, P_3\}$ as they are $A_3$'s two least favorite agents, the decentralized matching gives the outcome $(A_1, P_1), (A_2, P_2), (A_3, P_4), (A_4, P_3)$. Hence $A_3$ and $A_4$ are both strictly better off. Moreover, suppose there is a coordination mechanism among arms such

that each arm only accepts the most preferred agent. The decentralized matching gives the outcome $(A_1, P_4), (A_2, P_3), (A_3, P_1), (A_4, P_2)$, which is the arm-optimal stable matching.

## A.4 Comparison of multi-stage and single-stage matching

In this example, we show the gap between multi-stage welfare and single-stage welfare. Suppose there are four arms $\mathcal{A} = \{A_1, A_2, A_3, A_4\}$ and three agents $\mathcal{P} = \{P_1, P_2, P_3\}$. Agents have varied quotas: $q_1 = 2$ and $q_2 = q_3 = 1$. Arms' attributes are given by $v_1 = v_2 = v_3 = 2, v_4 = 1$, and $e_{13} = e_{23} = e_{32} = 0, e_{12} = e_{22} = e_{31} = 0.5, e_{11} = e_{21} = e_{33} = 1, e_{14} = 0.2, e_{24} = 0.5,$ $e_{34} = 0.8$. The latent utilities and arms' true preferences are shown in Table 1. Suppose each agent uses the straightforward strategy by pulling its most preferred arms up to the quota. Then the single-stage matching has the outcome $(A_1, P_1), (A_2, P_1), (A_3, P_3)$. The multi-stage matching gives the outcome $(A_1, P_1), (A_2, P_1), (A_3, P_3), (A_4, P_2)$. Hence $P_2$ is strictly better off in multi-stage matching as $P_2$'s welfare increases from 0 to 1.5 by changing from single-stage matching to multi-stage matching. On the other hand, $P_1$ and $P_3$ have the same welfare in single-stage and multi-stage matching. This result corroborates Proposition 3.

## A.5 Supplementary results for real application

We give supplementary results to the real data analysis, where the admission data is from the *New York Times* "The Choice" blog (available at https://thechoice.blogs.nytimes.com/category/admissions-data). Two colleges, Harvard and Yale, are excluded from the sample due to a significant proportion of missing values.

### A.5.1 Chi-squared test with FDR control

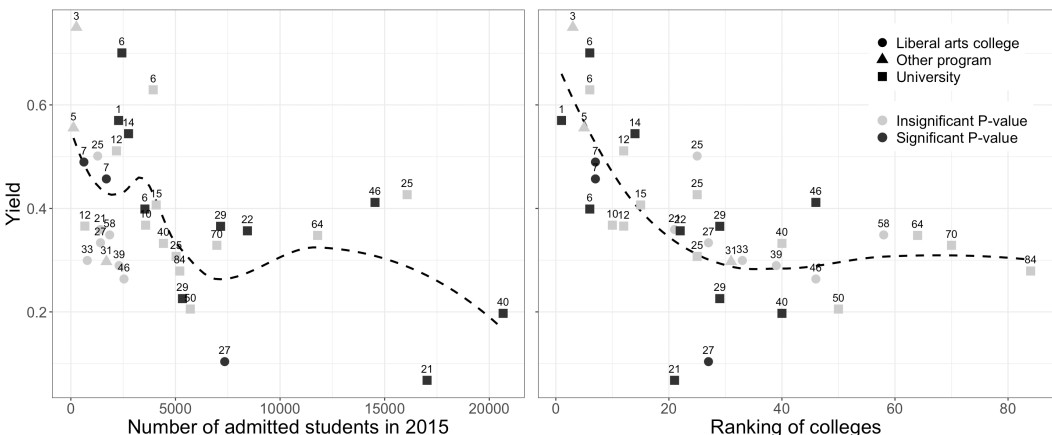

Figure 5: Regression of the yield on the size of admitted class and the ranking, respectively. We fit the dashed curves using smoothing splines with the tuning parameter chosen by GCV. The labels $\{1, 2, \dots, 35\}$ of each point indicates colleges' ranking according to *U.S. News and World Report*, where two (or more) colleges might tie in the ranking, and liberal arts colleges, national universities, and other undergraduate programs are ranked separately within their categories. Gray and black points denote colleges with insignificant and significant $p$-values, respectively, in chi-squared tests under an FDR control.

We test if the yields of colleges changed over 2015–17. The null hypothesis is that the state is the same. We use a simultaneous chi-squared test for all colleges with the count data on accepted and enrolled students and under an FDR control at a .05 significance level [9]. Figure 5 shows that colleges with large numbers of admitted students are likely to have significantly varied yields. Moreover, top-ranked national universities and liberal arts colleges are likely to have significantly varied yields. This observation corroborates the uncertainty in applicants' preferences facing colleges. Tables 3 and 4 report the 13 colleges with significant $p$-values and the 22 colleges with insignificant $p$-values, respectively.

Table 3: 13 chi-squared tests with significant $p$-value under the FDR control at the .05 significance level. Colleges' ranking data are from *U.S. News and World Report*. The "Y/N" means the use of the waiting list varied during 2015–17.

|  | $p$-value | Category | Ranking | Waiting list |
|---|---|---|---|---|
| Boston University | .0013 | National University | 40 | Yes |
| Brown University | .0012 | National University | 14 | No |
| Claremont McKenna College | .0003 | Liberal Arts College | 7 | Y/N |
| College of Holy Cross | $2.20E$-16 | Liberal Arts College | 27 | Yes |
| Emory University | $2.20E$-16 | National University | 21 | Yes |
| Georgia Tech | .0022 | National University | 29 | Yes |
| Middlebury College | .0065 | Liberal Arts College | 7 | Y/N |
| Princeton University | $8.31E$-12 | National University | 1 | Yes |
| Stanford University | $2.50E$-06 | National University | 6 | Y/N |
| University of Chicago | $2.20E$-06 | National University | 6 | Y/N |
| University of Rochester | .0001 | National University | 29 | Y/N |
| USC | $2.31E$-11 | National University | 22 | No |
| University of Wisconsin | .0008 | National University | 46 | Y/N |

Table 4: 22 chi-squared tests with insignificant $p$-value under the FDR control at the .05 significance level. Colleges' ranking data are from *U.S. News and World Report*. The "Y/N" means the use of the waiting list varied during 2015–17.

|  | $p$-value | Category | Ranking | Waiting list |
|---|---|---|---|---|
| Babson College | .8994 | Other Program | 31 | Yes |
| Barnard College | .6159 | Liberal Arts College | 25 | Yes |
| Bates College | .0798 | Liberal Arts College | 21 | Yes |
| CalTech | .0584 | National University | 12 | Y/N |
| Carnegie Mellon University | .4988 | National University | 25 | Yes |
| College of William&Mary | .2227 | National University | 40 | Yes |
| Cooper Union | .9512 | Other Program | 3 | Yes |
| Dartmouth College | .2217 | National University | 12 | Y/N |
| Dickinson College | .4727 | Liberal Arts College | 46 | Y/N |
| Elon University | .6872 | National University | 84 | Y/N |
| George Washington University | .0309 | National University | 70 | Yes |
| Johns Hopkins University | .1799 | National University | 10 | Yes |
| Kenyon College | .8012 | Liberal Arts College | 27 | Yes |
| Lafayette College | .8719 | Liberal Arts College | 39 | Yes |
| Olin College of Engineering | .5317 | Other Program | 5 | Y/N |
| Rensselaer Polytech | .0285 | National University | 50 | Y/N |
| Scripps College | .6511 | Liberal Arts College | 33 | Y/N |
| St. Lawrence University | .0587 | Liberal Arts College | 58 | Yes |
| University of Maryland | .4438 | National University | 64 | Y/N |
| University of Michigan | .0277 | National University | 25 | Y/N |
| University of Pennsylvania | .3665 | National University | 6 | Y/N |
| Vanderbilt University | .7576 | National University | 15 | Y/N |

### A.5.2 Evidence on hierarchical structure

We present the evidence on the hierarchical structure in the sense that students who were invited to the waiting list and remain available at a later stage are likely to be far worse than the admitted

students at the regular admission stage. The report of National Association for College Admission Counseling [29] shows that the admission rate of the waiting list is significantly lower than that of regular admission. The top students in a college's waiting list, uncertain about their rankings in the list and whether the college would admit them later, may have accepted offers from their less preferred colleges. We calculate the admission rate of the waiting list as follows:

$$\frac{\text{the number of offers sent to wait-listed students}}{\text{the total number of students invited to the waiting list}}.$$

Figure 6 reports that the majority ($> 77\%$) of admission rate of the waiting list are below $5\%$, which result corroborates the existence of the hierarchical structure in college admissions with waiting lists.

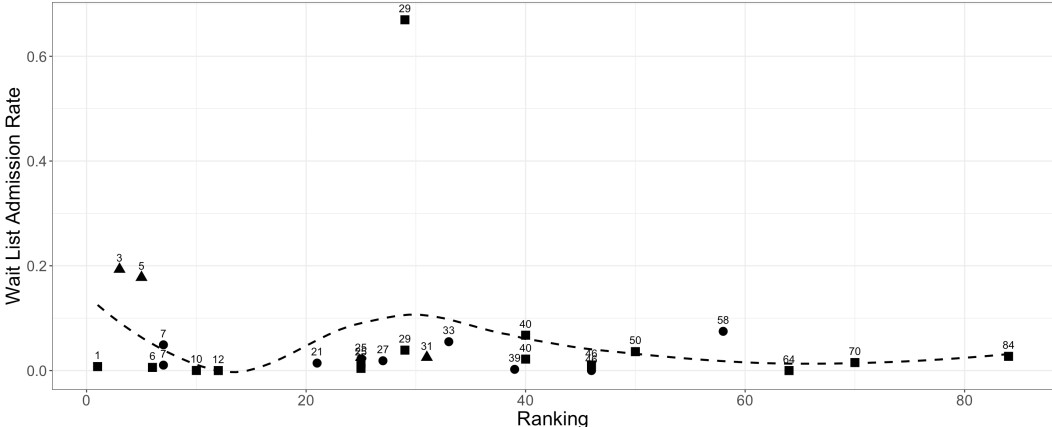

Figure 6: The regression uses smoothing splines with the tuning by GCV.

# B   Proofs

## B.1   Proof of Theorem 1

### B.1.1   Hierarchical structure

We the exploit the underlying hierarchical structure of the optimization problem in Eq. (3). For an arm set $\mathcal{B}_{i,k} \subseteq \{\mathcal{A}_k \setminus \cup_{l \leq k-1} \mathcal{B}_{i,l}\}$, its loss can be formulated by comparing its expected payoff to the expected payoff of $\bar{\mathcal{B}}_{i,k}$, where we suppose that $\cup_{k \in [K]} \bar{\mathcal{B}}_{i,k}$ achieves the optimal value $\bar{\mathcal{U}}_i$ in (3). Then the loss of $\mathcal{B}_{i,k}$ for any $k \in [K]$ becomes

$$\mathcal{L}_{i,k}[\mathcal{B}_{i,k}] = \mathbf{1}\Big\{ \sum_{j \in \mathcal{B}_{i,k}} \pi_{i,k}(s^*_{i,k}, v_j) > q_i - \text{card}(\cup_{l \leq k-1} \mathcal{C}_{i,l}) \Big\} \text{OE}[\mathcal{B}_{i,k}] \tag{B.11}$$
$$+ \mathbf{1}\Big\{ \sum_{j \in \mathcal{B}_{i,k}} \pi_{i,k}(s^*_{i,k}, v_j) \leq q_i - \text{card}(\cup_{l \leq k-1} \mathcal{C}_{i,l}) \Big\} \text{UE}[\mathcal{B}_{i,k}].$$

Here the *over-enrollment* (OE) loss in (B.11) is defined as

$$\text{OE}[\mathcal{B}_{i,k}] \equiv \gamma_i \Big\{ \sum_{j \in \mathcal{B}_{i,k}} \pi_{i,k}(s^*_{i,k}, v_j) + \text{card}(\cup_{l \leq k-1} \mathcal{C}_{i,l}) - q_i \Big\}$$
$$- \Big\{ \sum_{j \in \mathcal{B}_{i,k}} (v_j + e_{ij})\pi_{i,k}(s^*_{i,k}, v_j) - \sum_{j \in \bar{\mathcal{B}}_{i,k}} (v_j + e_{ij})\pi_{i,k}(s^*_{i,k}, v_j) \Big\}, \quad \forall k \in [K],$$

where we recall that penalty parameter $\gamma_i$ is defined in (2). The *under-enrollment* (UE) loss in (B.11) is given by

$$\text{UE}[\mathcal{B}_{i,k}]$$
$$\equiv \begin{cases} \rho_{i,k}[\sum_{j \in \bar{\mathcal{B}}_{i,k}} (v_j + e_{ij})\pi_{i,k}(s^*_{i,k}, v_j) - \sum_{j \in \mathcal{B}_{i,k}} (v_j + e_{ij})\pi_{i,k}(s^*_{i,k}, v_j)], & k \leq K-1, \\ \sum_{j \in \bar{\mathcal{B}}_{i,k}} (v_j + e_{ij})\pi_{i,K}(s^*_{i,k}, v_j) - \sum_{j \in \mathcal{B}_{i,k}} (v_j + e_{ij})\pi_{i,K}(s^*_{i,k}, v_j), & k = K, \end{cases} \tag{B.12}$$

where $\rho_{i,k} \in (0,1)$ is a discount factor for $k \leq K-1$. Note that $\rho_{i,k} < 1$ is because $P_i$ can fill the remaining quota (if any) in subsequent stages of the matching process. On the other hand, $\rho_{i,k} > 0$ is due to the observation that the arms available at subsequent stages are likely to be worse than the arms available at the current stage. Specifically, we refer to this observation as the *hierarchical structure* of the multi-stage matching and it is defined as follows: For any agent $P_i$, the $j$th best arm available at the subsequent stage has lower latent utility than the $j$th best arm available at the current stage, where $j \geq 1$. The hierarchical structure has been noted in college admissions with waiting lists [13]. Unlike the stages $k \leq K-1$, the last stage $k = K$ has the discount factor equals to 1 since the agent cannot fill the remaining quota (if any) after the last stage.

The formulation in Eq. (B.11) allows one to study stage-wise optimal sets $\mathcal{B}_{i,k}$ that minimize the loss $\mathcal{L}_{i,k}$ for each $k \in [K]$. This makes the optimization problem easier compared to jointly finding $\mathcal{B}_{i,k}$ for all $k \in [K]$ such that $\cup_{k \in [K]} \mathcal{B}_{i,k}$ maximizes the expected payoff in (3).

### B.1.2 Main proof of Theorem 1

*Proof.* We introduce additional notations. Let $V_{i,k}(s_{i,k}^*, \mathcal{B}_{i,k})$ be the expected utility of arms from $\mathcal{B}_{i,k} \subseteq \{\mathcal{A}_k \setminus \cup_{l \leq k-1} \mathcal{B}_{i,l}\}$ for agent $P_i$ at stage $k \in [K]$. That is,

$$V_{i,k}(s_{i,k}^*, \mathcal{B}_{i,k}) \equiv \sum_{j \in \mathcal{B}_{i,k}} (v_j + e_{ij}) \pi_{i,k}(s_{i,k}^*, v_j).$$

Let $\mathcal{N}_{i,k}(s_{i,k}^*, \mathcal{B}_{i,k})$ be the expected number of arms in $\mathcal{B}_{i,k}$ accepting $P_i$. That is,

$$\mathcal{N}_{i,k}(s_{i,k}^*, \mathcal{B}_{i,k}) \equiv \sum_{j \in \mathcal{B}_{i,k}} \pi_{i,k}(s_{i,k}^*, v_j).$$

By Lagrangian duality, the optimization of $\mathcal{L}_{i,k}[\mathcal{B}_{i,k}]$ in Eq. (B.11) can be reformulated to the constraint form:

$$\underbrace{\max_{\mathcal{B}_{i,k} \subseteq \{\mathcal{A}_k \setminus \cup_{l \leq k-1} \mathcal{B}_{i,l}\}} \left\{ V_{i,k}(s_{i,k}^*, \mathcal{B}_{i,k}) - \gamma_i \max\{\mathcal{N}_{i,k}(s_{i,k}^*, \mathcal{B}_{i,k}) + \mathrm{card}(\cup_{l \leq k-1} \mathcal{C}_{i,l}) - q_i, 0\} \right\}}_{\mathcal{I}_1},$$

$$\text{s.t. } \underbrace{\mathrm{UE}(\mathcal{B}_{i,k}) \geq \eta_{i,k}'}_{\mathcal{I}_2},$$

Here $\eta_{i,k}' > 0$ is an appropriately chosen tolerance parameter for $k \leq K-1$, and $\eta_{i,K}' = 0$. The constraint $\mathcal{I}_2$ can be written as

$$V_{i,k}(s_{i,k}^*, \mathcal{B}_{i,k}) \leq V_{i,k}(s_{i,k}^*, \mathcal{B}_{i,k}^*) - \eta_{i,k}', \quad \forall s_{i,k}^*, \tag{B.13}$$

where $\mathcal{B}_{i,k} \subseteq \{\mathcal{A}_k \setminus \cup_{l \leq k-1} \mathcal{B}_{i,l}\}$. Since $\pi_{i,k}(\cdot, \cdot)$ is assumed to belong to an RKHS, $\pi_{i,k}(\cdot, \cdot)$ is bounded [39]. By Hoeffding's bound, with probability at least $1 - e^{-\epsilon}, \forall \epsilon > 0$,

$$V_{i,k}(s_{i,k}^*, \mathcal{B}_{i,k}) < \mathbb{E}_{s_{i,k}^*}[V_{i,k}(s_{i,k}^*, \mathcal{B}_{i,k})] + \sqrt{2\epsilon \sum_{j \in \mathcal{B}_{i,k}} \delta_{i,k}^2(v_j)(v_j + e_{ij})^2}$$

$$< \mathbb{E}_{s_{i,k}^*}[V_{i,k}(s_{i,k}^*, \mathcal{B}_{i,k})] + \sqrt{2\epsilon} \sum_{j \in \mathcal{B}_{i,k}} \delta_{i,k}(v_j)(v_j + e_{ij}).$$

Hence a sufficient condition for Eq. (B.13) is to control

$$\sum_{j \in \mathcal{B}_{i,k}} \delta_{i,k}(v_j)(v_j + e_{ij}) < \eta_{i,k}'', \quad \text{for } \mathcal{B}_{i,k} \subseteq \{\mathcal{A}_k \setminus \cup_{l \leq k-1} \mathcal{B}_{i,l}\}. \tag{B.14}$$

Here $\eta_{i,k}'' > 0$ is a tolerance parameter for $k \leq K-1$. Both the $\mathcal{I}_1$ and Eq. (B.14) are convex, and so by Lagrangian duality, they can be reformulated in the penalized form that finding $\mathcal{B}_{i,k} \subseteq \{\mathcal{A}_k \setminus \cup_{l \leq k-1} \mathcal{B}_{i,l}\}$ to maximize

$$\sum_{j \in \mathcal{B}_{i,k}} (v_j + e_{ij})[\pi_{i,k}(s_{i,k}, v_j) - \eta_{i,k} \delta_{i,k}(v_j)]$$

$$- \gamma_i \max\{\mathcal{N}_{i,k}(s_{i,k}^*, \mathcal{B}_{i,k}) + \mathrm{card}(\cup_{l \leq k-1} \mathcal{C}_{i,l}) - q_i, 0\},$$

where $\eta_{i,k} > 0$ for $k \leq K-1$ and $\eta_{i,K} = 0$. This completes the proof. $\qquad \square$

### B.2 Proof of Theorem 2

#### B.2.1 Quantifying the cutoff for the greedy strategy

Let $b_{i,k}$ be the value of $r$ of those arms on the cutoff. That is, arms on the cutoff satisfy $b_{i,k} = (v+e_i)[1 - \eta_{i,k}\delta_{i,k}(v)\pi_{i,k}^{-1}(s_{i,k}, v)] \geq 0$. Let $\Pi_{i,k}(b_{i,k})$ be the expected number of arms in $\widehat{\mathcal{B}}_{i,k}(s_{i,k})$ that would accept $P_i$. That is,

$$\Pi_{i,k}(b_{i,k})$$
$$= \sum_{j \in \mathcal{A}} \mathbf{1}\left(e_{ij} \geq \min\left\{\max\left\{b_{i,k}[1 - \eta_{i,k}\delta_{i,k}(v_j)\pi_{i,k}^{-1}(s_{i,k}, v_j)]^{-1} - v_j, 0\right\}, 1\right\}\right)\pi_{i,k}(s_{i,k}, v_j).$$

If there exists some $b_{i,k} \geq 0$ such that $\Pi_{i,k}(b_{i,k}) = q_i - \mathrm{card}(\cup_{l \leq k-1}\mathcal{C}_{i,l})$, we let $\widehat{b}_{i,k}(s_{i,k}) = b_{i,k}$ and the cutoff $\widehat{e}_{i,k}(s_{i,k}, v) = \min\{\max\{\widehat{b}_{i,k}(s_{i,k})[1 - \eta_{i,k}\delta_{i,k}(v)\pi_{i,k}^{-1}(s_{i,k}, v)]^{-1} - v, 0\}, 1\}$. However, if there is no solution to $\Pi_{i,k}(b_{i,k}) = q_i - \mathrm{card}(\cup_{l \leq k-1}\mathcal{C}_{i,l})$, we let

$$b_{i,k}^+(s_{i,k}) = \underset{b_{i,k} \geq 0}{\arg\max}\left\{\Pi_{i,k}(b_{i,k}) > q_i - \mathrm{card}(\cup_{l \leq k-1}\mathcal{C}_{i,l})\right\},$$
$$b_{i,k}^-(s_{i,k}) = \underset{b_{i,k} \geq 0}{\arg\min}\left\{\Pi_{i,k}(b_{i,k}) < q_i - \mathrm{card}(\cup_{l \leq k-1}\mathcal{C}_{i,l})\right\}.$$

To choose between $b_{i,k}^+$ and $b_{i,k}^-$, it is necessary to balance the expected utility and the expected penalty for exceeding the quota due to pulling arms on the *boundary*. Define two cutoffs $e_{i,k}^+(s_{i,k}, v) \equiv \min\{\max\{b_{i,k}^+(s_{i,k})[1 - \eta_{i,k}\delta_{i,k}(v)\pi_{i,k}^{-1}(s_{i,k}, v)]^{-1} - v, 0\}, 1\}$ and $e_{i,k}^-(s_{i,k}, v) \equiv \min\{\max\{b_{i,k}^-(s_{i,k})[1 - \eta_{i,k}\delta_{i,k}(v)\pi_{i,k}^{-1}(s_{i,k}, v)]^{-1} - v, 0\}, 1\}$. The two cutoffs correspond to two sets, $\mathcal{B}_{i,k}^+(s_{i,k}) = \{j \mid e_{ij} \geq e_{i,k}^+(s_{i,k}, v_j)\}$ and $\mathcal{B}_{i,k}^-(s_{i,k}) = \{j \mid e_{ij} \geq e_{i,k}^-(s_{i,k}, v_j)\}$, respectively. Consider the following condition for the arms on the boundary $\{\mathcal{B}_{i,k}^+(s_{i,k}) \setminus \mathcal{B}_{i,k}^-(s_{i,k})\}$. This condition formalizes the comparison of the variational expected utility and the expected penalty of exceeding the quota:

$$\sum_{j \in \mathcal{B}_{i,k}^+(s_{i,k}) \setminus \mathcal{B}_{i,k}^-(s_{i,k})} (v_j + e_{ij})[\pi_{i,k}(s_{i,k}, v_j) - \eta_{i,k}\delta_{i,k}(v_j)]$$
$$\geq \gamma_i \sum_{j \in \mathcal{B}_{i,k}^+(s_{i,k})} \pi_{i,k}(s_{i,k}, v_j) - \gamma_i[q_i - \mathrm{card}(\cup_{l \leq k-1}\mathcal{C}_{i,l})]. \tag{B.15}$$

If (B.15) holds, let $\widehat{b}_{i,k}(s_{i,k}) = b_{i,k}^+(s_{i,k})$ and otherwise, let $\widehat{b}_{i,k}(s_{i,k}) = b_{i,k}^-(s_{i,k})$. Then the cutoff

$$\widehat{e}_{i,k}(s_{i,k}, v) = \min\left\{\max\left\{\widehat{b}_{i,k}(s_{i,k})[1 - \eta_{i,k}\delta_{i,k}(v)\pi_{i,k}^{-1}(s_{i,k}, v)]^{-1} - v, 0\right\}, 1\right\}. \tag{B.16}$$

Finally, using the greedy strategy, agent $P_i$ pulls arms from

$$\widehat{\mathcal{B}}_{i,k}(s_{i,k}) = \{j \mid A_j \in \{\mathcal{A}_k \setminus \cup_{l \leq k-1}\mathcal{B}_{i,l}\} \text{ with } (v_j, e_{ij}) \text{ satisfying } e_{ij} \geq \widehat{e}_{i,k}(s_{i,k}, v_j)\}$$
$$= \{j \mid A_j \in \{\mathcal{A}_k \setminus \cup_{l \leq k-1}\mathcal{B}_{i,l}\} \text{ satisfying } r(A_j) \geq r_*\},$$

where $r_*$ is the cutoff defined in Section 3.1.

#### B.2.2 Main proof of Theorem 2

*Proof.* We define the function,

$$\mathrm{UE}^\dagger \equiv \left[\min_{j \in \mathcal{B}_{i,k}^-(s_{i,k})}(v_j + e_{ij})(1 - \eta_{i,k}\pi_{i,k}^{-1}(s_{i,k}, v_j)\delta_{i,k}(v_jf))\right]$$
$$\cdot \left[q_i - \mathrm{card}(\cup_{l \leq k-1}\mathcal{C}_{i,l}) - \sum_{j \in \mathcal{B}_{i,k}^-(s_{i,k})}\pi_{i,k}(s_{i,k}, v_j)\right].$$

It is not hard to see that $\mathrm{UE}^\dagger \geq 0$ and it equals 0 if there is a continuum of arms and $\pi_{i,k}(\cdot, v)$ is continuous in $v$. We divide the main proof of Theorem 2 into five steps.

**Step 1.** We show that the optimal strategy prefers an arm with higher fit given the same score. Suppose that arms $A_{j_1}, A_{j_2} \in \{\mathcal{A}_k \setminus \cup_{l \leq k-1} \mathcal{B}_{i,l}\}$ have the same score $v_{j_1} = v_{j_2}$, but $A_{j_1}$ has a worse fit than $A_{j_2}$ to agent $P_i$. Now assume that $A_{j_1}$ was pulled by $P_i$ at stage $k$ but $A_{j_2}$ was not, that is, $A_{j_1} \in \widehat{\mathcal{B}}_{i,k}(s_{i,k}), A_{j_2} \notin \widehat{\mathcal{B}}_{i,k}(s_{i,k})$. Then the expected number of arms accepting $P_i$ is unchanged if $P_i$ replaces $A_{j_1}$ with $A_{j_2}$ in $\widehat{\mathcal{B}}_{i,k}(s_{i,k})$. On the other hand, since the loss function in Eq. (4) is strictly decreasing in fit $e_{ij}$, $P_i$ should pull $A_{j_2}$ instead $A_{j_1}$. This argument holds regardless of strategies of other agents.

**Step 2.** We show that the cutoff curve $\widehat{e}_{i,k}(s_{i,k}, v)$ in Eq. (B.16) is well-defined. If the boundary $\{\mathcal{B}_{i,k}^+(s_{i,k}) \setminus \mathcal{B}_{i,k}^-(s_{i,k})\}$ is not empty, then $P_i$ pulling an arm $A_j$ on the boundary yields the loss

$$\mathcal{L}_{i,k}^\dagger[A_j] \leq 0,$$

which justifies the condition specified by Eq. (B.15). Since $\widehat{e}_{i,k}(s_{i,k}, v) \in [0, 1]$, the cutoff curve is well-defined.

**Step 3.** We show that the cutoff strategy of pulling arms from the set $\widehat{\mathcal{B}}_{i,k}(s_{i,k})$ is near-optimal. Let $\widetilde{\mathcal{B}}_{i,k}(s_{i,k})$ be any other arm set. Define the following mixed strategy:

$$\sigma_{i,k}(s_{i,k}, v, e_i; t) \equiv t \cdot \mathbf{1}\{(v, e_i) \in \widetilde{\mathcal{B}}_{i,k}(s_{i,k})\} + (1-t) \cdot \mathbf{1}\{(v, e_i) \in \widehat{\mathcal{B}}_{i,k}(s_{i,k})\}, \quad \text{for } t \in [0, 1].$$

The corresponding loss of the mixed strategy $\sigma_i$ is

$$
\bar{\mathcal{L}}_{i,k}(t) = \sum_{j \in \{\mathcal{A}_k \setminus \cup_{l \leq k-1} \mathcal{B}_{i,l}\}} (v_j + e_{ij})[\eta_{i,k}\delta_{i,k}(v_j) - \pi_{i,k}(s_{i,k}, v_j)]\sigma_{i,k}(s_{i,k}, v_j, e_{ij}; t)
$$
$$
+ \gamma_i \max\left\{ \sum_{j \in \{\mathcal{A}_k \setminus \cup_{l \leq k-1} \mathcal{B}_{i,l}\}} \pi_{i,k}(s_{i,k}, v_j)\sigma_{i,k}(s_{i,k}, v_j, e_{ij}; t) + \text{card}(\cup_{l \leq k-1}\mathcal{C}_{i,l}) - q_i, 0 \right\}.
$$

It is clear that $\bar{\mathcal{L}}_{i,k}(t)$ is convex in $t$. We discuss the local change $d\bar{\mathcal{L}}_{i,k}(0)/dt$ in three cases.

Case (I): Consider removing a single arm from $\widehat{\mathcal{B}}_{i,k}(s_{i,k})$. If the arm is from the non-empty boundary $\{\mathcal{B}_{i,k}^+(s_{i,k}) \setminus \mathcal{B}_{i,k}^-(s_{i,k})\}$, the condition specified by Eq. (B.15) implies that the loss $\bar{\mathcal{L}}_{i,k}(t)$ increases if not pulling the arm. Moreover, by construction, any other arm $A_j$ in $\widehat{\mathcal{B}}_{i,k}(s_{i,k})$ satisfies

$$(v_j + e_{ij})[\pi_{i,k}(s_{i,k}^*, v_j) - \eta_{i,k}\delta_{i,k}(v_j)] > \widehat{b}_{i,k}(s_{i,k})\pi_{i,k}(s_{i,k}^*, v_j)$$
$$\geq \gamma_i \sum_{j' \in \widehat{\mathcal{B}}_{i,k}(s_{i,k})} \pi_{i,k}(s_{i,k}, v_{j'}) - \gamma_i[q_i - \text{card}(\cup_{l \leq k-1}\mathcal{C}_{i,l})].$$

Hence, removing $A_j$ from $\widehat{\mathcal{B}}_{i,k}(s_{i,k})$ results in a strict increase in $\bar{\mathcal{L}}_{i,k}(t)$. We have $d\bar{\mathcal{L}}_{i,k}(0)/dt > 0$ in this case. By the convexity of $\bar{\mathcal{L}}_{i,k}(t)$ in $t$, we obtain

$$\bar{\mathcal{L}}_{i,k}(1) = \bar{\mathcal{L}}_{i,k}(0) + \frac{d\bar{\mathcal{L}}_{i,k}(0)}{dt}(1 - 0) > \bar{\mathcal{L}}_{i,k}(0),$$

Case (II): Consider adding a new arm with attributes $\{v_{j'}, e_{ij'}\}$ to $\widehat{\mathcal{B}}_{i,k}(s_{i,k})$, where the new arm is not from the set $\mathcal{B}_{i,k}^+(s_{i,k})$. Denote by $\mathcal{B}_{i,k}'(s_{i,k})$ the new arm set with the added arm. Note that $P_i$ pulls a new arm only if the arm reduces the loss $\bar{\mathcal{L}}_{i,k}(t)$, that is,

$$(v_{j'} + e_{ij'})[\pi_{i,k}(s_{i,k}, v_{j'}) - \eta_{i,k}\delta_{i,k}(v_{j'})]$$
$$\geq \gamma_i \sum_{j \in \mathcal{B}_{i,k}'(s_{i,k})} \pi_{i,k}(s_{i,k}, v_j) - \gamma_i[q_i - \text{card}(\cup_{l \leq k-1}\mathcal{C}_{i,l})]. \tag{B.17}$$

Since the added new arm is not in $\mathcal{B}^+_{i,k}(s_{i,k})$ and $\sum_{j \in \mathcal{B}^+_{i,k}(s_{i,k})} \pi_{i,k}(s_{i,k}, v_j) \geq q_i - \text{card}(\cup_{l \leq k-1} \mathcal{C}_{i,l})$, we have

$$
\begin{aligned}
\sum_{j \in \mathcal{B}'_{i,k}(s_{i,k})} &\pi_{i,k}(s_{i,k}, v_j) - [q_i - \text{card}(\cup_{l \leq k-1} \mathcal{C}_{i,l})] \\
&\geq \sum_{j \in \mathcal{B}'_{i,k}(s_{i,k})} \pi_{i,k}(s_{i,k}, v_j) - \sum_{j \in \mathcal{B}^+_{i,k}(s_{i,k})} \pi_{i,k}(s_{i,k}, v_j) \\
&\geq \pi_{i,k}(s_{i,k}, v_{j'}) \\
&\geq \pi_{i,k}(s_{i,k}, v_{j'}) - \eta_{i,k} \delta_{i,k}(v_{j'}).
\end{aligned}
\tag{B.18}
$$

Because that $\gamma_i > \sup_{j \in \mathcal{A}} \{v_j + e_{ij}\}$ and $\eta_{i,k} \geq 0$, the result in Eq. (B.18) is contradictory to Eq. (B.17). Hence, adding a new arm to $\widehat{\mathcal{B}}_{i,k}(s_{i,k})$ results in an increase in the loss $\bar{\mathcal{L}}_{i,k}(t)$. Hence, $d\bar{\mathcal{L}}_{i,k}(0)/dt > 0$ in this case. By the convexity of $\bar{\mathcal{L}}_{i,k}(t)$ in $t$, we obtain

$$
\bar{\mathcal{L}}_{i,k}(1) = \bar{\mathcal{L}}_{i,k}(0) + \frac{d\bar{\mathcal{L}}_{i,k}(0)}{dt}(1-0) > \bar{\mathcal{L}}_{i,k}(0),
$$

Case (III): Consider removing an arm with attributes $(v_j, e_{ij})$ from $\widehat{\mathcal{B}}_{i,k}(s_{i,k})$ and simultaneously adding new arms to $\widehat{\mathcal{B}}_{i,k}(s_{i,k})$. Suppose that the new arms have attributes $(v_{j''}, e_{ij''})$ and are from $\mathcal{B}''_{i,k}(s_{i,k})$. If $\widehat{\mathcal{B}}_{i,k}(s_{i,k}) = \mathcal{B}^-_{i,k}(s_{i,k})$, then the new arms are not in $\mathcal{B}^-_{i,k}(s_{i,k})$ and by definition,

$$
\begin{aligned}
(v_{j''} + e_{ij''})&[\pi_{i,k}(s_{i,k}, v_{j''}) - \eta_{i,k}\delta_{i,k}(v_{j''})]\pi^{-1}_{i,k}(s_{i,k}, v_{j''}) \\
&\leq \min_{j \in \mathcal{B}^-_{i,k}(s_{i,k})} \left\{ (v_j + e_{ij})[\pi_{i,k}(s_i, v_j) - \eta_i \delta_{i,k}(v_j)]\pi^{-1}_{i,k}(s_{i,k}, v_j) \right\}.
\end{aligned}
$$

Hence,

$$
\begin{aligned}
\mathcal{L}^\dagger_i&[\mathcal{B}^-_{i,k}(s_{i,k})] - \bar{\mathcal{L}}_{i,k}(1) \\
&\leq \sum_{j'' \in \mathcal{B}''_{i,k}(s_{i,k})} (v_{j''} + e_{ij''})[\pi_{i,k}(s_{i,k}, v_{j''}) - \eta_{i,k}\delta_{i,k}(v_{j''})]\pi^{-1}_{i,k}(s_{i,k}, v_{j''}) \cdot \pi_{i,k}(s_{i,k}, v_{j''}) \\
&\leq \left[ \min_{j \in \mathcal{B}^-_{i,k}(s_{i,k})} (v_j + e_{ij})(1 - \eta_{i,k}\pi^{-1}_{i,k}(s_{i,k}, v_j)\delta_{i,k}(v_j)) \right] \\
&\qquad \cdot \left[ q_i - \text{card}(\cup_{l \leq k-1}\mathcal{C}_{i,l}) - \sum_{j \in \mathcal{B}^-_{i,k}(s_{i,k})} \pi_{i,k}(s_{i,k}, v_j) \right]
\end{aligned}
\tag{B.19}
$$

$$
= \text{UE}^\dagger.
$$

If $\widehat{\mathcal{B}}_{i,k}(s_{i,k}) = \mathcal{B}^+_{i,k}(s_{i,k})$, then by definition of $\mathcal{B}^+_{i,k}(s_{i,k})$

$$
\mathcal{L}^\dagger_i[\mathcal{B}^+_{i,k}(s_{i,k})] - \bar{\mathcal{L}}_{i,k}(1) \leq \mathcal{L}^\dagger_i[\mathcal{B}^-_{i,k}(s_{i,k})] - \bar{\mathcal{L}}_{i,k}(1) \leq \text{UE}^\dagger.
$$

where the last inequality is by Eq. (B.19). Hence,

$$
\bar{\mathcal{L}}_{i,k}(0) - \bar{\mathcal{L}}_{i,k}(1) \leq \text{UE}^\dagger.
$$

Therefore, exchanging an arm in $\widehat{\mathcal{B}}_{i,k}(s_{i,k})$ with arms not in $\widehat{\mathcal{B}}_{i,k}(s_{i,k})$ could result in an increase in the loss $\bar{\mathcal{L}}_{i,k}(t)$ by at most $\text{UE}^\dagger$. Combining the cases (I), (II), (III), we obtain that

$$
\mathcal{L}^\dagger_i[\widehat{\mathcal{B}}_{i,k}(s_{i,k})] \leq \min_{\mathcal{B}_{i,k} \subseteq \{\mathcal{A}_k \setminus \cup_{l \leq k-1}\mathcal{B}_{i,l}\}} \mathcal{L}^\dagger_i[\mathcal{B}_{i,k}] + \text{UE}^\dagger.
$$

**Step 4.** We prove the other direction of the inequality. Since $\widehat{\mathcal{B}}_{i,k}(s_{i,k}) \subseteq \{\mathcal{A}_k \setminus \cup_{l \leq k-1}\mathcal{B}_{i,l}\}$,

$$
\mathcal{L}^\dagger_i[\widehat{\mathcal{B}}_{i,k}(s_{i,k})] \geq \min_{\mathcal{B}_{i,k} \subseteq \{\mathcal{A}_k \setminus \cup_{l \leq k-1}\mathcal{B}_{i,l}\}} \mathcal{L}^\dagger_i[\mathcal{B}_{i,k}].
$$

**Step 5.** If there is a continuum of arms and $\pi_{i,k}(\cdot, v)$ is continuous in $v$, then there exists $b_{i,k} \geq 0$ such that $\Pi_{i,k}(b_{i,k}) = q_i - \text{card}(\cup_{l \leq k-1} \mathcal{C}_{i,l})$, where $\Pi_{i,k}(b_{i,k})$ is defined in Section 3.1:

$$\Pi_{i,k}(b_{i,k}) = \sum_{j \in \mathcal{A}} \mathbf{1} \left( e_{ij} \geq \min \left\{ \max \left\{ b_{i,k}[1 - \eta_{i,k}\delta_{i,k}(v_j)\pi_{i,k}^{-1}(s_{i,k}, v_j)]^{-1} - v_j, 0 \right\}, 1 \right\} \right) \pi_{i,k}(s_{i,k}, v_j).$$

Therefore, by definition, $\widehat{\mathcal{B}}_{i,k}(s_{i,k}) = \mathcal{B}_{i,k}^+(s_{i,k}) = \mathcal{B}_{i,k}^-(s_{i,k})$, and

$$q_i - \text{card}(\cup_{l \leq k-1}\mathcal{C}_{i,l}) - \sum_{j \in \mathcal{B}_{i,k}^-(s_{i,k})} \pi_{i,k}(s_{i,k}, v_j) = 0.$$

Hence $\text{UE}^\dagger = 0$. This completes the proof. $\qquad\square$

### B.3 Proof of Theorem 3

#### B.3.1 Main proof of Theorem 3

*Proof.* We follow the proof arguments for Theorem 4 in [15]. The only difference is that here we define the following penalized expected utility and the expected number of arms:

$$V_{i,k}(s_{i,k}^*, \widehat{\mathcal{B}}_{i,k}) \equiv \sum_{j \in \widehat{\mathcal{B}}_{i,k}} (v_j + e_{ij})[\pi_{i,k}(s_{i,k}^*, v_j) - \eta_{i,k}\delta_{i,k}(v_j)],$$

$$\mathcal{N}_{i,k}(s_{i,k}^*, \widehat{\mathcal{B}}_{i,k}) \equiv \sum_{j \in \widehat{\mathcal{B}}_{i,k}} \pi_{i,k}(s_{i,k}^*, v_j).$$

We omit the details for simplicity. $\qquad\square$

#### B.3.2 Calibration under the worst-case loss

Besides the average-case loss in Theorem 3, we also consider the worst-case loss with respect to the unknown $s_{i,k}^*$. Theorem 4 gives *minimax calibration*, which calibrates $s_{i,k}$ to minimize the maximum loss $\max_{s_{i,k}^*}\{\mathcal{L}_{i,k}^\dagger[\widehat{\mathcal{B}}_{i,k}(s_{i,k})]\}$ over the unknown $s_{i,k}^*$.

**Theorem 4.** *The worse-case loss* $\max_{s_i^*}\{\mathcal{L}_{i,k}^\dagger[\widehat{\mathcal{B}}_{i,k}(s_{i,k})]\}$ *is minimized if* $s_{i,k} \in [0, 1]$ *is chosen as the solution to*

$$\sum_{j \in \widehat{\mathcal{B}}_{i,k}(s_{i,k})} 2(v_j + e_{ij})\delta_{i,k}(v_j) + \sum_{j \in \widehat{\mathcal{B}}_{i,k}(0)} (v_j + e_{ij})\left[\pi_{i,k}(0, v_j) - \eta_{i,k}\delta_{i,k}(v_j)\right]$$

$$= \sum_{j \in \widehat{\mathcal{B}}_{i,k}(1)} (v_j + e_{ij})\left[\pi_{i,k}(1, v_j) - \eta_{i,k}\delta_{i,k}(v_j)\right] + \gamma_i \sum_{j \in \widehat{\mathcal{B}}_{i,k}(s_{i,k})} \pi_{i,k}(1, v_j) - \gamma_i q_i.$$

The Proof follows from Theorem 5 in [15].

### B.4 Proof of Proposition 1

*Proof.* Recall the cutoff parameter $\widehat{b}_{i,k}(s_{i,k})$ defined in Eq. (B.16). Similarly, we define a cutoff parameter $b'_{i,k}(s_{i,k})$ for the linear cutoff: $e'_{i,k}(s_{i,k}, v) = \min\{\max\{b'_{i,k}(s_{i,k}) - v, 0\}, 1\}$ following three steps. First, we define that

$$\Pi'_{i,k}(b_{i,k}) \equiv \sum_{j \in \mathcal{A}} \mathbf{1}(e_{ij} \geq \min\{\max\{b_{i,k} - v_j, 0\}, 1\})\pi_{i,k}(s_{i,k}, v_j).$$

If there exists $b_{i,k} \geq 0$ such that $\Pi'_{i,k}(b_{i,k}) = q_i - \text{card}(\cup_{l \leq k-1}\mathcal{C}_{i,l})$, we let $b'_{i,k}(s_{i,k}) = b_{i,k}$. Second, if there is no solution to $\Pi'_{i,k}(b_{i,k}) = q_i - \text{card}(\cup_{l \leq k-1}\mathcal{C}_{i,l})$, we let

$$b_{i,k}^+(s_{i,k}) = \arg\max_{b_{i,k} \geq 0} \left\{\Pi'_{i,k}(b_{i,k}) > q_i - \text{card}(\cup_{l \leq k-1}\mathcal{C}_{i,l})\right\},$$

$$b_{i,k}^-(s_{i,k}) = \arg\min_{b_{i,k} \geq 0} \left\{\Pi'_{i,k}(b_{i,k}) < q_i - \text{card}(\cup_{l \leq k-1}\mathcal{C}_{i,l})\right\}.$$

Define that
$$e_{i,k}^+(s_{i,k}, v) \equiv \min\{\max\{b_{i,k}^+(s_{i,k}) - v, 0\}, 1\},$$
and
$$e_{i,k}^-(s_{i,k}, v) \equiv \min\{\max\{b_{i,k}^-(s_{i,k}) - v, 0\}, 1\}.$$

Then $e_{i,k}^+(s_{i,k}, v)$ and $e_{i,k}^-$ correspond to following sets respectively,
$$\mathcal{B}_{i,k}^+(s_{i,k}) = \{j \mid e_{ij} \geq e_{i,k}^+(s_{i,k}, v_j)\},$$
and
$$\mathcal{B}_{i,k}^-(s_{i,k}) = \{j \mid e_{ij} \geq e_{i,k}^-(s_{i,k}, v_j)\}.$$

Consider the following condition for the arms on the boundary $\{\mathcal{B}_{i,k}^+(s_{i,k}) \setminus \mathcal{B}_{i,k}^-(s_{i,k})\}$:
$$\sum_{j \in \mathcal{B}_{i,k}^+(s_{i,k}) \setminus \mathcal{B}_{i,k}^-(s_{i,k})} (v_j + e_{ij})\pi_{i,k}(s_{i,k}, v_j) \geq \gamma_i \sum_{j \in \mathcal{B}_{i,k}^+(s_{i,k})} \pi_{i,k}(s_{i,k}, v_j) - \gamma_i q_i.$$

If the above condition holds, let $b_{i,k}'(s_{i,k}) = b_{i,k}^+(s_{i,k})$ and otherwise, let $b_{i,k}'(s_{i,k}) = b_{i,k}^-(s_{i,k})$. Third, we let the linear cutoff
$$e_{i,k}'(s_{i,k}, v) = \min\{\max\{b_{i,k}'(s_{i,k}) - v, 0\}, 1\}.$$

Now for any $\eta_{i,k} \geq 0$ and state $s_{i,k}$, the set of arms that have justified envy is
$$\mathcal{V}(\eta_{i,k}, s_{i,k}) = \left\{ (v, e_i) \ \middle| \ \frac{\widehat{b}_{i,k}(s_{i,k})}{1 - \eta_{i,k}\delta_{i,k}(v)\pi_{i,k}^{-1}(s_{i,k}, v)} > v + e_i > b_{i,k}'(s_{i,k}) \right\}.$$

Hence the probability that an arm with attributes $(v, e)$ has justified envy is increasing in
$$\frac{\widehat{b}_{i,k}(s_{i,k})}{1 - \eta_{i,k}\delta_{i,k}(v)\pi_{i,k}^{-1}(s_{i,k}, v)}. \tag{B.20}$$

Note that (B.20) is strictly increasing in the arm's uncertainty level $\delta_{i,k}(v)\pi_{i,k}^{-1}(s_{i,k}, v)$, the probability that an arm has justified envy is strictly increasing in the arm's uncertainty level. $\qquad\square$

## B.5 Proof of Proposition 2

*Proof.* Adopting the proof in Section B.4, we note that the number of arms having justified envy is
$$\sum_{j \in \{\mathcal{A}_k^{T+1} \setminus \cup_{l \leq k-1}\mathcal{B}_{i,l}\} \cap \mathcal{V}(\eta_i, s)} \left[ \frac{\widehat{b}_{i,k}(s_{i,k})}{1 - \eta_{i,k}\delta_{i,k}(v_j)\pi_{i,k}^{-1}(s_{i,k}, v_j)} - b_{i,k}'(s_{i,k}) \right]. \tag{B.21}$$

The term in the bracket of Eq. (B.21), i.e.,
$$\frac{\widehat{b}_{i,k}(s_{i,k})}{1 - \eta_{i,k}\delta_{i,k}(v_j)\pi_{i,k}^{-1}(s_{i,k}, v_j)} - b_{i,k}'(s_{i,k})$$

is strictly increasing in $\eta_{i,k}$. Hence the number of arms having justified envy is strictly increasing in $\eta_{i,k} \geq 0$. This completes the proof. $\qquad\square$

## B.6 Proof of Proposition 3

*Proof.* We show the improved welfare for agents by construction. Consider the strategy of an agent, for example, $P_i$ with $i \in [m]$. Suppose that $P_i$ pulls arms at the first stage in multi-stage matching using the strategy that $P_i$ would have used in single-stage matching. All arms that would have accepted $P_i$ in single-stage matching accept $P_i$. The reason is that arms have incomplete information on what other offers are coming in later stages. Hence, $P_i$ can achieve at least as well as its payoff from single-stage matching. Therefore, agents benefit from multi-stage matching. $\qquad\square$

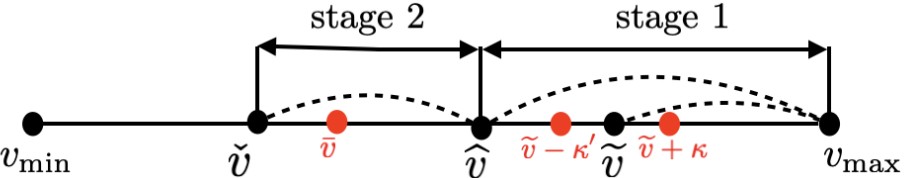

Figure 7: Cutoffs at the two stages.

## B.7 Proof of Proposition A.4

*Proof.* First, we consider the matching outcome of the straightforward strategy by pulling arms according to the latent utilities. Suppose that agents $P_1$ and $P_2$ use the CDM algorithm, which is a straightforward strategy and calibrates the uncertain state in the same way as LUB-CDM [15]. The calibration in Theorem 3 calibrates the state parameters as $s = s_a$ for $P_1$ and $s = s_b$ for $P_2$. We note that worst-case calibration in Theorem 4 gives the same calibrations in this example. Thus, $P_1$ and $P_2$ pull the same set of arms at the first stage, where the arms' scores $v \geq \widetilde{v}$ and the cutoff $\widetilde{v}$ satisfies

$$\sum_{j \in \mathcal{A}} \mathbf{1}(v_j \geq \widetilde{v}) \cdot s_a \cdot (1 - p^*) = q, \quad \text{and} \quad \sum_{j \in \mathcal{A}} \mathbf{1}(v_j \geq \widetilde{v}) \cdot (1 - s_b) \cdot (1 - p^*) = q. \quad \text{(B.22)}$$

Here the boundary arm set is assumed to be empty in Eq. (B.22). Next, we consider $P_3$'s strategy. Arms with the scores worse than $\widetilde{v}$ will accept $P_3$ since if they accept $P_3$, they get $u_3$ for sure, but if they reject $P_3$, they will at best be pulled by $P_1$ or $P_2$ with probability $(1 - p^*)$ and get the utility at most $u_1$, but $u_3 > (1 - p^*)u_1$. Suppose now $P_3$ pulls arms with the score $v \geq \widehat{v}$, where $\widehat{v} < \widetilde{v}$. By Eq. (B.22), there are total $(p^*)^2 q [s_a (1 - p^*)]^{-1}$ of arms with $v \geq \widetilde{v}$ that are not pulled by $P_1$ or $P_2$ and they will accept $P_3$. Thus, we can quantify $\widehat{v}$ by letting it satisfy

$$\sum_{j \in \mathcal{A}} \mathbf{1}(\widehat{v} \leq v_j < \widetilde{v}) = q \left[ 1 - \frac{(p^*)^2}{s_a(1 - p^*)} \right].$$

See an illustration of the cutoffs in Figure 7. Then we analyze $P_1$'s expected payoff by using the CDM. If the true state is $s_b$, $P_1$ does not fill its capacity during the first stage and needs to pull more arms at the second stage. Suppose that $P_1$ pulls arms with $v \in [\check{v}, \widehat{v})$ at the second stage, where $\check{v}$ satisfies

$$\sum_{j \in \mathcal{A}} \mathbf{1}(\check{v} \leq v_j < \widehat{v}) \cdot (1 - p^*) = q - \sum_{j \in \mathcal{A}} \mathbf{1}(v_j \geq \widetilde{v}) \cdot s_b \cdot (1 - p^*). \quad \text{(B.23)}$$

Hence, $P_1$'s expected payoff by using CDM is

$$\mathcal{U}_1^{\text{CDM}} = \frac{1}{2}(1 - p^*) \left[ \sum_{v_j \geq \widetilde{v}} v_j + \sum_{\check{v} \leq v_j < \widehat{v}} v_j \right]. \quad \text{(B.24)}$$

We then consider the matching outcome of the LUB-CDM algorithm. Suppose that $P_1$ uses the LUB-CDM while $P_2$ still uses the CDM. By Theorem 2, $P_1$ pulls arms according to the ranking of the following quantity:

$$v_j \left[ 1 - \eta_{1,1} \cdot \frac{\delta_{1,1}(v_j)}{s} \right] = v_j \left[ 1 - \eta_{1,1} \cdot \frac{s_a - s_b}{2s} \right], \quad \text{(B.25)}$$

where $\delta_{1,1}(v) = \frac{1}{2}(s_a - s_b)$ in this example and $\eta_{1,1} \geq 0$ is the regularization parameter defined in Theorem 1. The calibration in Theorem 3 calibrates the state parameter as $s = s_a$ for $P_1$. Then $P_1$ pulls the arms with the score $v \in [\widetilde{v} - \kappa', \widetilde{v}) \cup \{v \geq \widetilde{v} + \kappa\}$ and rejects those with $v \in [\widetilde{v}, \widetilde{v} + \kappa)$. Here the boundary arm set is assumed to be empty, and $\kappa, \kappa'$ satisfy

$$\sum_{j \in \mathcal{A}} \mathbf{1}(\widetilde{v} - \kappa' \leq v_j < \widetilde{v}) = \sum_{j \in \mathcal{A}} \mathbf{1}(\widetilde{v} \leq v_j < \widetilde{v} + \kappa) \cdot s_a. \quad \text{(B.26)}$$

By Eq. (B.25), $\kappa$ and $\kappa'$ also need to satisfy that

$$\widetilde{v} - \kappa' = (\widetilde{v} + \kappa) \left[ 1 - \eta_{1,1} \cdot \frac{s_a - s_b}{2s_a} \right]. \quad \text{(B.27)}$$

Then we analyze $P_1$'s expected payoff by using the LUB-CDM. If the true state is $s_b$, $P_1$ needs to pull more arms at the second stage. Since the second stage is the last stage and by Theorem 1, it is optimal for $P_1$ to choose $\eta_{1,2} = 0$, where the LUB-CDM coincides with the CDM. Suppose that $P_1$ pulls arms with $v \in [\bar{v}, \hat{v}]$ at the second stage, where $\bar{v}$ satisfies

$$
\begin{aligned}
&\sum_{j \in \mathcal{A}} \mathbf{1}(\bar{v} \leq v_j < \hat{v}) \cdot (1 - p^*) \\
&= q - \sum_{j \in \mathcal{A}} \mathbf{1}(v_j \geq \widetilde{v} + \kappa) \cdot s_b \cdot (1 - p^*) - \sum_{j \in \mathcal{A}} \mathbf{1}(\widetilde{v} - \kappa' \leq v_j < \widetilde{v}) \cdot (1 - p^*).
\end{aligned}
\tag{B.28}
$$

Subtracting Eq. (B.28) from Eq. (B.23), we obtain that

$$
\begin{aligned}
\sum_{j \in \mathcal{A}} \mathbf{1}(\check{v} \leq v_j < \bar{v}) &= \sum_{j \in \mathcal{A}} \mathbf{1}(\widetilde{v} - \kappa' \leq v_j < \widetilde{v}) - \sum_{j \in \mathcal{A}} \mathbf{1}(\widetilde{v} \leq v_j < \widetilde{v} + \kappa) \cdot s_b \\
&= \sum_{j \in \mathcal{A}} \mathbf{1}(\widetilde{v} \leq v_j < \widetilde{v} + \kappa) \cdot (s_a - s_b) > 0.
\end{aligned}
\tag{B.29}
$$

where the second equality is by Eq. (B.26). Thus, $\bar{v} > \check{v}$, and the $P_1$'s expected payoff by using the LUB-CDM is

$$
\mathcal{U}_1^{\text{LUB-CDM}} = (1 - p^*) \sum_{\widetilde{v} - \kappa' \leq v_j < \widetilde{v}} v_j + \frac{1}{2}(1 - p^*) \left[ \sum_{v_j \geq \widetilde{v} + \kappa} v_j + \sum_{\bar{v} \leq v_j < \hat{v}} v_j \right].
\tag{B.30}
$$

We now comparing the two expected payoffs in Eqs. (B.30) and (B.24), respective. By taking the difference, we have

$$
\begin{aligned}
&\mathcal{U}_1^{\text{LUB-CDM}} - \mathcal{U}_1^{\text{CDM}} \\
&= (1 - p^*) \sum_{\widetilde{v} - \kappa' \leq v_j < \widetilde{v}} v_j - \frac{1}{2}(1 - p^*) \left[ \sum_{\widetilde{v} \leq v_j < \widetilde{v} + \kappa} v_j + \sum_{\check{v} \leq v_j < \bar{v}} v_j \right] \\
&> (1 - p^*)(\widetilde{v} - \kappa') \sum_{j \in \mathcal{A}} \mathbf{1}(\widetilde{v} - \kappa' \leq v_j < \widetilde{v}) \\
&\quad - (\widetilde{v} + \kappa) \sum_{j \in \mathcal{A}} \mathbf{1}(\widetilde{v} \leq v < \widetilde{v} + \kappa) - \bar{v} \sum_{j \in \mathcal{A}} \mathbf{1}(\check{v} \leq v < \bar{v}) \\
&= [(\widetilde{v} - \bar{v})(s_a - s_b) - (2\kappa' s_a + \kappa)] \sum_{j \in \mathcal{A}} \mathbf{1}(\widetilde{v} \leq v_j < \widetilde{v} + \kappa) \\
&= \{(s_a - s_b)[(1 - \eta_{1,1})\widetilde{v} - \bar{v}] + [2s_a - \eta_{1,1}(s_a - s_b) - 1]\kappa\} \sum_{j \in \mathcal{A}} \mathbf{1}(\widetilde{v} \leq v_j < \widetilde{v} + \kappa),
\end{aligned}
\tag{B.31}
$$

where the second equality is due to Eqs. (B.26) and (B.29), and the last equality is by Eq. (B.27). For sufficiently small $\kappa$ and $\eta_{1,1}$, we have

$$
\mathcal{U}_1^{\text{LUB-CDM}} > \mathcal{U}_1^{\text{CDM}}.
$$

Last, we quantify the improvement of the expected payoff. From Eq. (B.26), $\kappa$ satisfies that

$$
\sum_{j \in \mathcal{A}} \mathbf{1}\left( (\widetilde{v} + \kappa)\left(1 - \eta_{1,1}\frac{s_a - s_b}{2s_a}\right) \leq v_j < \widetilde{v} \right) = \sum_{j \in \mathcal{A}} \mathbf{1}(\widetilde{v} \leq v_j < \widetilde{v} + \kappa) \cdot s_a.
$$

Suppose that $v_j$ is uniformly distributed, we have a first-order approximation of the above equations:

$$
\widetilde{v} - (\widetilde{v} + \kappa)\left(1 - \eta_{1,1}\frac{s_a - s_b}{2s_a}\right) = \kappa s_a,
$$

which implies that

$$
\kappa = \frac{\widetilde{v}(s_a - s_b)\eta_{1,1}}{2s_a(1 + s_a) - (s_a - s_b)\eta_{1,1}} = O(\eta_{1,1})
$$

Plugging it to Eq. (B.31) suggests that a sufficient condition for $\mathcal{U}_1^{\text{LUB-CDM}} > \mathcal{U}_1^{\text{CDM}}$ is

$$\eta_{1,1} < \frac{2s_a}{s_a - s_b} \cdot \frac{(1 + s_a)(s_a - s_b)(\widetilde{v} - \bar{v})}{(s_a - s_b)(\widetilde{v} - \bar{v}) + (2s_a^2 + 1)\widetilde{v}}. \tag{B.32}$$

By the condition that $\kappa' > 0$, we have

$$\eta_{1,1} < \frac{2s_a}{s_a - s_b}(1 + s_a - \widetilde{v}). \tag{B.33}$$

Under Eqs. (B.32) and (B.33), and noting that,

$$\sum_{j \in \mathcal{A}} \mathbf{1}(\widetilde{v} \leq v_j < \widetilde{v} + \kappa) = O(\kappa) = O(\eta_{1,1}),$$

we have that,

$$\mathcal{U}_1^{\text{LUB-CDM}} - \mathcal{U}_1^{\text{CDM}} = O(\eta_{1,1}).$$

This completes the proof. $\qquad\qquad\qquad\qquad\qquad\qquad\qquad\qquad\qquad\qquad\qquad\qquad$ $\square$