# OpenReview forum: "Learning in Multi-Stage Decentralized Matching Markets"
_NeurIPS.cc/2021/Conference — NeurIPS 2021 Poster_

### Official Review · Reviewer_GLYK · 2021-07-15

**Rating:** 7
**Confidence:** 4

**Summary:**

The paper studies learning in multi-stage decentralized markets. Motivated by applications such as college admissions and the academic job market, the paper studies the classic matching problem under the assumptions that agents do not know their preferences and matching decisions can take place in multiple stages. As in the classic matching problem, a constraint is that colleges and applicants have quotas (e.g., students cannot accept more than one offer).

The paper models this problem as a multi-armed bandit problem with K>1 stages and quota constraints. Colleges are the agents and students are the arms. The arms’ preferences have no restrictions and are modeled through a probability function. Each agent values both a common quality aspect (the “score”) and a “fit” which is agent-specific. In this model, the paper proposes an algorithm that is based on the notions of lower uncertainty bound and calibrated decentralized matching; this algorithm learns the optimal strategy at each stage. Further, the paper investigates the economic implications of the model. For example, they show that strategic behavior may lead to unfairness (measured in terms of no justified envy) and that agents are better off with multi-stage compared to single-stage matching. Finally, they include simulations using data from college admissions.


**Limitations And Societal Impact:**

The authors should discuss the technical limitations of their model, discuss/motivate their assumptions and empirical analysis. This discussion is currently not included in the paper.

**Main Review:**

The paper contributes to the growing literature in the intersection of learning and matching markets. The main novelty of the question (compared to the existing literature) is the assumption of multiple stages. Due to the similarity of the question with previous works, the technical/ algorithmic contribution is not particularly interesting. However, matching happening in multiple stages is a real concern in practice and the authors provide several insights about the implications of their findings.

Main comments:
- A common problem in college admissions is the lack of fully rational (but still strategic) behavior and access to full information on the student’s side, which makes the modeling on students’ side less realistic. How does the model address these concerns?
The related literature is incomplete. Several recent papers study very related questions, e.g.,
 Liu et al. (2020),    "Bandit Learning in Decentralized Matching Markets,"
 Liu et  al. (2020),   "Competing Bandits in Matching Markets,"
  Immorlica et al. (2020), "Information Acquisition in Matching Markets: The Role of Price Discovery."

Furthermore, the paper is very closely related to [13]. How do the two papers compare from a modeling and technical perspective? What is the novelty of the current paper? The comparison to this work and the papers above should be made clear.

- Does assuming just 2 stages ($K=2$) -- which would be realistic for college admissions -- help get more specific insights about the model?
- How would the results change if arms behaved similarly to agents? E.g., instead of assuming some acceptance probability function, their decisions had more structure (like the agents’ decisions).
- Lines 124-126: Assuming negligible application costs is not realistic. In fact, it would be interesting to study how relaxing this assumption would affect the results.
- I didn’t find the result about the hierarchical structure in Theorem 1 particularly surprising but it was nice to see this intuition formalized.
- The paper uses very heavy notation. For the sake of readability, I believe that some technical details and definitions could be omitted (for example, in Section 3.2).
- I enjoyed the results in Section 4. It’s worth exploring the economic implications of the model further.
- Although the notion of no justified envy is pretty standard, I think it’s quite misleading as a measure of fairness. It would be better to explicitly use the term “no justified envy” instead of “fairness”.
- In Proposition 3: Is there any characterization of the gap possible? It would be nice to study this question (at least empirically).
- The presentation in Section 5 needs improvement. It is unclear what information the data contained and how the authors used it in the simulations.
- The insight that universities are better off by employing strategic admissions (see lines 350-351) seems to be true for less selective colleges. Top colleges tend to admit the best (but potentially earlier than others) so I am not sure how this finding reflects reality.



**Time Spent Reviewing:**

5.5

---

> ### Author Response · Authors · 2021-08-10
> **Response to Reviewer GLYK**
>
> Thank you for your careful evaluations and constructive suggestions. In the following, your specific comments are listed, followed by our responses.
>
> [Comment 1] A common problem in college admissions is the lack of fully rational (but still strategic) behavior and access to full information on the students’ side, which makes the modeling on students’ side less realistic. How does the model address these concerns?
>
> Response: Our model in Eq. (2) of Section 2 addresses these two concerns. First, for addressing the concern of lacking fully rational behavior on the students’ side, our model explicitly considers the uncertainty of students’ preferences. That is, the students in the current year may have different preferences compared to the students in the past year. The uncertainty of students’ preferences could be due to the variability in the relative popularity of colleges over time. We incorporate the uncertainty of preferences into the state parameter $s_{i,k}$.
>
> Second, for addressing the concern of lacking full information on the students’ side, our model assumes that a college does not have any knowledge on a student except their score and fit. For example, a college in our model has no information on how other colleges would value a student and how a student would rank the colleges. In our model, the acceptance probability $\pi_{i,k}(s_{i,k},v_j)$ incorporates the colleges’ competition into the dependence on the score $v_j$.
>
>
>
> [Comment 2] The related literature is incomplete. Several recent papers study very related questions, e.g., Liu et al. (2020a), “Bandit Learning in Decentralized Matching Markets,” Liu et al. (2020b), “Competing Bandits in Matching Markets,” Immorlica et al. (2020), “Information Acquisition in Matching Markets: The Role of Price Discovery.” Furthermore, the paper is very closed related to [13] (i.e., Dai and Jordan, 2020). How do the two papers compare from a modeling and technical perspective? What is the novelty of the current paper?
>
> Response: Thanks for the references. We compare our paper with these recent papers as follows.
>
> First, there exists a significant difference between the multi-stage decentralized matching problem in our paper and the multi-armed bandit problem in Liu et al. (2020a, b). A bandit problem is a sequential allocation problem in which an environment repeatedly provides an agent with a fixed set of arms. Although similarly involving sequential decision-making under limited information, the multi-stage matching market has multiple agents competing for arms, and an arm exits the market once it accepts an offer. The competition induces a hierarchical structure, making the optimization in multi-stage decentralized matching different from that in multi-armed bandits.
>
> Second, Immorlica et al. (2020) studied the model of college admissions that accounts for students’ information acquisition costs in forming their preferences. In contrast, we consider that students know their preferences, which are unknown to colleges. Moreover, the students’ preferences are uncertain due to competition among colleges and variability in the relative popularity of colleges over time. Our setting is motivated by the real data discussed in Section 5.
>
> Third, Dai and Jordan (2020) studied single-stage matching, which is fundamentally different from multi-stage matching. The novelty of the current paper can be summarized into three parts—
> 	(i) New modeling. This paper incorporates the hierarchical structure arising from the multi-stage matching into the decision-making. We employ variational analysis to model the optimization under the hierarchical structure given in Theorem 1.
> 	(ii) New theoretical results. Unlike single-stage matching, we find that the optimal solution in multi-stage matching involves an additional step of estimating uncertainty bounds.
> 	(iii) New economic implications. Different from single-stage matching, there exists a welfare-versus-fairness trade-off in multi-stage matching. Participants will strategically act in favor of a low uncertainty level of acceptance to reduce competition and increase the expected payoff.
>
> [Comment 3] Does assuming just 2 stages (K = 2) — which would be realistic for college admissions — help get more specific insights about the model?
>
> Response: We give an example of 2 stages (K=2) of college admission, such as regular admission and waiting list admission. By Theorem 1 and Proposition 2, colleges in the first stage (i.e., regular admission) would strategically send offers to students with low uncertainty levels (i.e., the probability of accepting the offers has a low uncertainty). In this way, colleges would reduce competition. In the second stage  (i.e., waiting list admission), colleges would act according to their true preferences. That is, colleges will admit the best wait-listed students.
>
>
> [Comment 4] How would the results change if arms behaved similarly to agents? E.g., instead of assuming some acceptance probability function, their decisions had more structure (like the agents’ decisions).
>
> Response: Our current analysis is motivated by college admission. The goal is to have colleges (i.e., agents) learning optimal strategies that maximize expected payoffs based on historical data. However, our algorithm can predict match compatibility for both agents and arms. Each arm can learn how much an agent may like them by predicting the probability that they can receive an offer from a specific agent. Then arms would make the decisions based on the prediction that if they have a realistic potential of receiving a better offer. We would like to leave this problem for future studies with appropriate real data.
>
>
> [Comment 5] Lines 124-126: Assuming negligible application costs is not realistic. In fact, it would be interesting to study how relaxing this assumption would affect the result.
>
> Response:  The assumption of negligible application costs follows the literature, e.g., Che and Koh (2016, J. Political Econ). It is conceivable that our algorithm could be extended to deal with non-negligible application costs, in which different colleges would have different sets of students applicants. This is because with non-negligible application costs, submitting applications to all colleges is not the dominant strategy for students (Avery and Levin, 2010, Am Econ Rev). However, once the student sets are given for each college, Algorithm 1 would still be applicable for finding the optimal strategies. On the other hand, treating non-negligible application costs would be more technical and tedious both in theory and simulation, and we shall leave them for future studies.
>
>
> [Comment 6] Although the notion of no justified envy is pretty standard, I think it’s quite misleading as a measure of fairness. I would be better to explicitly use the term “no justified envy” instead of “fairness”.
>
> Response:  We will change the writing and explicitly use the term “no justified envy” whenever possible in the paper. On the other hand, we adopted the notion of no justified envy as a measure of fairness by following the literature; see, e.g., Che and Koh (2016, J. Political Econ).
>
>
> [Comment 7]: In Proposition 3: Is there any characterization of the gap possible? It would be nice to study this question (at least empirically).
>
> Response: We will provide an empirical example to illustrate the gap between multi-stage welfare and single-stage welfare. Consider the numerical example in Section S1.3 in Supplementary Appendix. We study the arms' latent utilities and preferences give in Table S1. Suppose each agent uses the straightforward strategy by pulling its most preferred arms up to the quota. Then the single-stage matching has the outcome $(A_1,P_1), (A_2,P_1), (A_3,P_3)$. On the other hand, the multi-stage matching gives the outcome $(A_1,P_1), (A_2,P_1), (A_3,P_3), (A_4,P_2)$. Hence $P_2$ is strictly better off in multi-stage matching as $P_2$'s welfare increases from $0$ to $1.5$ by switching from single-stage matching to multi-stage matching. On the other hand, ​$P_1$ and $P_2$ have the same outcome in both single-stage and multi-stage matching.  This result corroborates Proposition 3.
>
>
> [Comment 8]: The presentation in Section 5 needs improvement. It is unclear what information the data contained and how the authors used it in the simulations.
>
> Response: We will add more details on the data and discuss how we used it in the simulations. The training data are simulated from colleges’ random proposing by pulling a random number of arms according to the latent utilities. The training data consists of $20$ times of random proposing under each of the arms' preference structures with the two-stage admissions. This training data simulates the graduate school admissions over $20$ years.  The testing data draws a random state from $\{s_1,\ldots,s_{10}\}$, which gives the corresponding arms' preferences. Then we apply Algorithm 1 with hyperparameters $\eta_{i,1}=0.1,\eta_{i,2} = 0$ and $\gamma_i=2.5$.
>
>
> [Comment 9]: The insight that university are better off by employing strategic admissions seems to be true for less selective colleges. Top colleges tend to admit the best (but potentially earlier than others) so I am not sure how this finding reflects reality.
>
> Response: This finding is obtained from the real data on 37 U.S. colleges admissions for 2015-17 applicants. The insight that top colleges are better off by employing strategic admissions also agrees with the simulation result in Figure 1 (left plot).

---

> > ### Comment · Reviewer_GLYK · 2021-08-24
> > **Thank you for the response and updated score**
> >
> > I would like to thank the authors for the great effort they put in their response. Conditional on implementing the suggested revisions, I found the response to most of my comments convincing enough. As a result, I have updated my score from 6 to 7.

---

> > > ### Author Response · Authors · 2021-08-27
> > > **Thank Reviewer GLYK for the Feedback**
> > >
> > > We want to thank the reviewer for the positive feedback and thank the reviewer for updating the score.

---

### Official Review · Reviewer_wCZe · 2021-07-16

**Rating:** 7
**Confidence:** 4

**Summary:**

This paper develops a nonparametric statistical framework to learn optimal strategies in multi-stage decentralized matching markets, where agents with limited capacity are matched with arms with uncertain preferences in several stages. They propose an algorithm relying on the concepts of lower uncertainty bound and calibrated decentralized matching to maximize agents’ expected payoff. The algorithm uses a data-driven approach to calibrate the arms’ uncertainty and penalize agents for exceeding capacity. They show several implications on welfare and fairness based on their model. First, they show that agents favor arms with low uncertainty in the levels of acceptance to maximize their welfare, leading to unfair outcomes for arms since they might not be picked by their favorite agents even though those rank below them are picked. Then, they show that agents do better in a multi-stage matching instead of a centralized one. They demonstrate the effectiveness of their algorithms through experiments on synthetic and college admissions data.

**Limitations And Societal Impact:**

The authors have adequately addressed the limitations and potential negative societal impact of their work.

**Main Review:**

The work uses a novel variational formulation and framework to obtain the optimal strategy in a multi-stage decentralized matching market. The claims are well supported by theoretical proofs and some experimental results on synthetic and college admission data. The writing was clear and concise, although more explanation and background on the theoretical parts would be nice. The results shed some light on fairness in the college admission process, which is an interesting result on its own. They built upon the concepts of lower uncertainty bound to compute each arm's expected utility and uncertainty and calibrated decentralized matching to calibrate each agent's quota and each arm's parameter on each stage depending on historical data. Their framework is flexible in allowing the use of various machine learning methods to estimate arms' expected utility and uncertainty bounds. They also provide convincing experimental results to show the effectiveness of their algorithm.

**Time Spent Reviewing:**

3

---

> ### Author Response · Authors · 2021-08-10
> **Response to Reviewer wCZe**
>
> Thank you for your careful evaluations and positive feedback. Following your suggestion, we will add more explanation and background on the theoretical parts.
>
> In particular, we will add more explanation on the choice of $\eta_{i,k}$ as follows. The parameter $\eta_{i,k}>0$ is induced by the hierarchical structure of arms in multi-stage matching. Here, the arms available at subsequent stages are worse than those available at the current stage. For the optimization problem in Section 3, the $\eta_{i,k}$ serves as a regularization parameter for the uncertainty measure $\delta_{i,k}$. In practice, we may choose a large value of $\eta_{i,k}$ if the agents’ competition is tense. This is because the arms left at subsequent stages are much worse than those available at the current stage.

---

### Official Review · Reviewer_zGHc · 2021-07-17

**Rating:** 6
**Confidence:** 3

**Summary:**

The paper studies matching in decentralized markets. A collection of agents (schools) and a collection of arms (students) form the two sides of the market. The matching is done over stages where agents select a collection of arms, an arm would choose only one agent with some probability and once matched would exit the market. Further, the agents have quotas which they cannot exceed. An important point is that the market is decentralized and that the probabilities are not known in advance and hence they have to be estimated. The paper thus involves both a learning step and a maximization step. Some theoretical characterizations of the fairness and welfare are also obtained.

**Limitations And Societal Impact:**

Limitations And Societal Impact are adequately addressed.

**Main Review:**

-I think the paper addresses an important and interesting topic and I do not think there has been much work in decentralized matching.

-In lines (155,181) NP-completeness is claimed. I do not see how the presented arguments are a valid justification. Isn't NP-completeness proved through a reduction not loosely by saying there are many possibilities to check. I think either a proof should be given or the claim should be that the proposed method would be computationally intractable.

-There is a large literature on stochastic matching and online matching which does not seem to be referred to at all. Further, it is possible that the techniques/results from stochastic/online matching could be useful here. See e.g., Mehta, "Online matching and ad allocation." (2013).

-Perhaps, the essential parts where the paper departs from most of the work in matching is decentralization and learning. With regards to the learning part, the inclusion of probabilities similar to \pi in the model have been considered before in online matching where an agent would be matched with an arm and the match succeeds with some probability. In these online matching settings, it is assumed that these probabilities are learned from previous records similar to this work. But the learning step is abstracted away. Why can't we abstract the learning step here as well? Can't we assume that we have the probabilities or noisy estimates of them. Further, algorithm 1 solves for the values B_{i,k} before the matching process begins, so this isn't an interactive learning setting, is it?


-Some parts of the paper do not seem new and have been used before in reference [13], e.g. the utility model and theorem 3.

-Although it would complicate things, it seems that the model would be more accurate if it were to allow the arms to change their minds in the next round and possibly drop the agent to which they were matched in favor of a better agent. Excluding this setting, does the algorithm have stability guarantees, i.e. is the final matching a stable matching?

-Why doesn't \pi_{i,k} also have a dependence on the fit e_{i,j}. Further, shouldn't \pi_{i,k} also have an index for j?

-Lines161,233: shouldn't the min and max arguments be over (i,k) instead of s_{i,k}




**Time Spent Reviewing:**

2

---

> ### Author Response · Authors · 2021-08-10
> **Response to Reviewer zGHc**
>
> Thank you for your careful evaluations and constructive suggestions. In the following, your specific comments are listed, followed by our responses.
>
>
>
> [Comment 1] In lines (155, 181) NP-completeness is claimed. I do not see how the presented arguments are a valid justification. Isn’t NP-completeness proved through a reduction not loosely by saying there are many possibilities to check. I think either a proof should be given or the claim should be that the proposed method would be computationally intractable.
>
> Response: We adopted the use of NP-completeness by following the literature; see, e.g., Section 2 of Milgrom (2017, Discovering prices. Columbia University Press). However, for clarity, we will change the writing in lines (155, 181) from NP-completeness to computational intractable.
>
>
>
> [Comment 2] There is a large literature on stochastic matching and online matching which does not seem to be referred to at all. Further, it is possible that the techniques/results from stochastic/online matching could be useful here. See e.g., Mehta, “Online matching and ad allocation.” (2013).
>
> Response: Thanks for the references. We compare the decentralized matching with stochastic/online matching, which has a rich history in studying the problem of finding a maximum matching in a graph and its generalization (e.g., Mehta, 2013). There exist two main differences. First, the agents in decentralized matching would compete for the arms with uncertain preferences. Here an agent's goal is maximizing welfare. This is different from the problem of finding a matching with the largest size in maximum matching literature. Second, the agents in decentralized matching make decisions independent of others, which is also different from the decision problem in maximum matching.
>
>
>
> [Comment 3] With regards to the learning part, the inclusion of probabilities similar to $\pi$ in the model have been considered before in online matching where an agent would be matched with an arm and the match succeeds with some probability. In these online matching settings, it is assumed that these probabilities are learned from previous records similar to this work. But the learning step is abstracted away. Why can’t we abstract the learning step here as well? Can’t we assume that we have probabilities or noisy estimates of them.
>
> Response: There exist many differences between the learning in our setting and the learning in online matching. First, we consider the uncertain arms’ preferences. The agents’ decisions are influenced by latent state variables whose values must be inferred in practice. This presents calibration challenges that need to take into account capacity limits and opportunity costs, which are different from the learning in online matching. Second, we consider multi-stage matching. The optimization problem in the multi-stage matching induces a hierarchical structure. Its variational problem requires learning a lower uncertainty bound, which is also different from the online matching literature.
>
> These learning elements are new and beyond simply assuming that we have noisy estimates.
>
>
>
> [Comment 4] Further, algorithm 1 solves for the value $B_{i,k}$ before the matching process begins, so this isn’t an interactive learning setting, is it?
>
> Response:  This is an interactive learning setting since there exist interactions between matching and learning. For example, at stage $k$, Algorithm 1 produces the set $B_{i,k}^L(s_{i,k})$ that contains arms for agent $P_i$ to pull. If some arms accept the offers at stage $k$ (i.e., being matched with agents), they exit the market. Next, at stage $k+1$, Algorithm 1 would produce a different set $B_{i,k+1}^L(s_{i,k+1})$ that contains arms for agent $P_i$ to pull. Hence this is an interactive setting between matching and learning.
>
>
>
> [Comment 5] Some parts of the paper do not seem new and have been used before in reference [13], e.g., the utility model and theorem 3.
>
> Response:  We note that reference [13] (i.e., Dai and Jordan, 2020) studied single-stage matching, which is fundamentally different from multi-stage matching. The novelty of the current paper can be summarized into three parts.
>
> (i) New model. Although employing a similar utility model, this paper incorporates the new hierarchical structure arising from the multi-stage matching into the decision-making. We use variational analysis to derive a new optimization model under the hierarchical structure, as given in Theorem 1.
>
> (ii) New theoretical results. Unlike single-stage matching, we find that solving the optimal strategy in multi-stage matching requires an additional step of estimating uncertainty bounds besides the calibration in Theorem 3.
>
> (iii) New economic implications. Different from single-stage matching, there exists a welfare-versus-fairness trade-off in multi-stage matching. Participants will strategically act in favor of a low uncertainty level of acceptance to reduce competition and increase the expected payoff in multi-stage matching.
>
>
>
> [Comment 6] Although it would complicate things, it seems that the model would be more accurate if it were to allow arms to change their minds in the next round and possibly drop the agent to which they were matched in favor of a better agent. Excluding this setting, does the algorithm have stability guarantees, i.e., is the final matching a stable matching?
>
> Response: We would like to clarify that the arms cannot change their minds in decentralized matching markets. For example, suppose a student accepts an offer in college admissions. In that case, she/he is typically not allowed to change their mind if later their favorite college sends them offers.
>
> Algorithm 1 generally does not have a stability guarantee. We show in Appendix Section S1.3 that Algorithm 1 may yield a different matching outcome compared to the deferred acceptance (DA) algorithm, where DA enjoys a stability guarantee. However, we would like to explain that, in general, it is challenging to guarantee stability in decentralized matching. This is mainly due to two reasons—
>
> (i) There is no centralized clearinghouse for coordinating agents in decentralized markets. Moreover, agents have no direct means of communication. For example, in college admissions, each college makes its decision independently of others’ decisions. As a result, each college has little information on other colleges’ preferences and decisions. Since the stability constraint requires knowing other agents’ preferences, it is difficult to incorporate the stability constraint into the design of an algorithm for decentralized matching without a centralized system. In contrast, the DA algorithm enjoys a stability guarantee partly due to clearinghouses in centralized matching markets.
>
> (ii) We have shown in Appendix S1.2—S1.3 that some agents are better off by using Algorithm 1 compared to stable matching outcomes. In many decentralized matching markets such as college admissions, agents have little incentive to sacrifice welfare for achieving stable matching.
>
>
>
> [Comment 7] Why doesn’t $\pi_{i,k}$ also have a dependence on the fit $e_{ij}$? Further, shouldn’t $\pi_{i,k}$ also have an index for j?
>
> Response:  We answer these two questions separately. First, the probability $\pi_{i,k}$ does not depend on $e_{ij}$ because $e_{ij}$ is a private value that is only known to the agent $P_i$. The arm $A_j$ does not know $e_{ij}$ and thus whether $A_j$ accepting $P_i$ does not depend on $e_{ij}$.
>
> Second, the probability $\pi_{i,k}$ does not have an index for $j$ because $\pi_{i,k}$ models the an arm's acceptance probability through the dependence on the score $v_j$ rather than the individual $A_j$; that is, $\pi_{i,k} = \pi_{i,k}(s_{i,k},v_j)$. In this way, we can learn $\pi_{i,k}$ using the historical data which contains arms with similar scores as $v_j$.
>
>
>
> [Comment 8] Line 161, 233: shouldn’t the min and max arguments be over $(i,k)$ instead of $s_{i,k}$?
>
> Response:  We would like to clarify that the min and max arguments should be over s_{i,k}; that is, these arguments in the current paper are correct. The reason is that for any given pair $(i,k)$, the $\delta_{i,k}$ in Line 161 measures the uncertainty of the acceptance probability with respect to $s_{i,k}$. A similar reason applies to $\widehat{\delta}_{i,k}$ in Line 233.

---

### Official Review · Reviewer_auj5 · 2021-07-20

**Rating:** 8
**Confidence:** 3

**Summary:**

The authors study a multi sided matching market under a statistical model, with the goal of finding the optimal strategies. The model uses agents and arms, but due to its applications the intuition is easier to capture assuming the agents are universities and the arms students. Specifically, there are $n$ agents and $m$ arms, which are pulled over $K$ rounds. Each agent has a quota $q_i \ge 1$ of arms they can use, and the sum of all quotas is at most $n$. The market is two-sided because the arms (students) have preferences over the agents (universities) (and these preferences can be fuzzy) and the agents (universities) have preferences over the arms (students): each arm $j$ has an intrinsic value $v_j$ and an extra value $e_{ij}$, specific to agent $i$.

At every step, each agent can pull any number of arms. However, each arm can only respond to at most one agent, or reject all of them. The arms don’t have such explicit preferences: their interaction with the agents is captures by the probability $\pi_{i,k}(s_{i,k}, v_j)$ that arm $i$ will accept agent $j$ at time $k$, where $s_{i,k}$ is the state at time $k$. This probability is generally not known but is inferred from data. Overall, agent $i$’s utility is the sum of $(v_j + e_{ij}) \cdot \pi_{i,k}(s_{i,k}, v_j) $ over every turn $k$ and all arms they tried using, minus a penalty if the quota $q_i$ is exceeded. Clearly, this induces a game between the agents, which have to decide how to use their quotas to select arms that are good enough, but not so good that they might be snatched by other agents.

The authors provide a variety of results:
1. Three algorithms (one of which is their main result) on the side of learning the optimal strategy from the perspective of agent $i$.
2. Fairness and Welfare trade-offs.
3. Experiments.

In particular, the first algorithm yields an optimal strategy for maximising the welfare of agent $i$ by minimising their *variational loss*, which is (roughly described) a modification of their utility with an added factor depending on $\max_{s_{i,k}} \pi_{i,k}(s_{i,k}, v_j)  - \min_{s_{i,k}} \pi_{i,k}(s_{i,k}, v_j)$ (the ‘variance’) and weighted by $\eta_{i, k}$, which reflects that subsequent rounds are typically less valuable than the first ones. Intuitively this makes sense: minimising the variance would maximise the way the quota is used. This strategy is NP-hard to solve for though, which leads to the development of a *greedy* approximation: rather than immediately solving for variance, the arms are chosen greedily based on their expected utility per unit of acceptance probability. The second algorithm (and arguably main result of the paper) uses historic data to estimate the $\pi_{i,k}(s_{i,k}, v_j)$’s. The actual algorithm has many details, but I hope is fairly represent an abstraction as follows:
1. Use an off the shelf estimator for the $\pi_{i,k}(s_{i,k}, v_j)$.
2. Modify them by subtracting a weighted variational term (as in the first algorithm), or set to 1 if the arm has not been used.
3. Balance exploration and exploitation of the arms used.

The fairness results revolve around *justified envy*: in a nutshell, an arm would have justified envy if it is pulled by agent $i’$, but agent $i$ (which the arm prefers) is trying a different arm on the same turn. This would reflect the real world situation where highly skilled individual might miss an offer because a potential employer might think they have a high chance of rejecting it. The authors show that higher uncertainly results in higher probability of an arm being envious. Moreover, the higher the $\eta_{i,k}$ terms are, the higher the number of envious arms. Finally, they compare with the single-stage matching markets (which do not produce envy in their optimal outcome) and show that multi-stage markets provide increased welfare, at the expense of fairness.

The authors also have experiments on synthetic data that support their learning procedure, comparing to a simpler strategy, where every agent selects their favourite arms within their remaining quota, at every step. As expected, both strategies perform well for popular agents, but have very different performance for less desirable ones. Again, this reflects that minimising variance (which is easier for popular agents: imagine a top university offering positions to students, knowing most would accept) is the key, which is further supported by extrapolating from real data sets of admissions and university rankings.


**Limitations And Societal Impact:**

The authors have adequately addressed the limitations and societal impact of this work.

**Main Review:**

The paper is written at a very high mathematician standard and a lot of effort has been put into fitting it within the constraints of the format: nevertheless, it is choked by the page limit and I still required multiple passes to make sure I understood the notation and finer points of the setting. The related work is very clearly exposed and addressed whenever necessary: there’s no doubt that the authors understand the field very well. My only comment on the presentation is, if possible, to add a short explanation on the choice of $\eta_{i, k}$: it is a crucial parameter that affects nearly every result, but I couldn’t find much more than “it exists” in the main text, without resorting to the Appendix for further details.

The model makes sense and the abstractions taken to make it workable mathematically are not too detached from reality: my main concern (perhaps as a game theorist) is if there was a way to more explicitly expose the competition aspect between agents. This is encoded through the $\pi_$’s and $s_{i, k}$: I am wonder how different a model where each arm has a randomly produced weak ordering of arms would be.

The fairness results are interesting, but I might have chosen a different definition of envy for this setting: restricting the envy to be ‘per turn’ simplified the analysis but begs the question: if an arm is passed over by its favourite agent and then pulled by less desirable one over two rounds, should it not have envy? Moreover, the arms might not know exactly what the strategy of the agents is. Perhaps it would be more natural to define envy ex-post, for the case where we have pairs (arm1, agent1) and (arm2, agent2), even though arm1 prefers agent2 over agent1 and agent2 prefers arm1 over arm2.

Overall, I enjoyed reading the paper, which studies a realistic model and produces impactful, punchy results which reassure our initial understanding and provide finer details as well.

**Time Spent Reviewing:**

5

---

> ### Author Response · Authors · 2021-08-10
> **Response to Reviewer auj5**
>
> Thank you for your careful evaluations and nice comments.  In the following, your specific comments are listed, followed by our responses.
>
>
> [Comment 1] My only comment on the presentation is, if possible, to add a short explanation on the choice of $\eta_{i,k}$.
>
> Response: Following your suggestion, we will add the following explanation on the choice of $\eta_{i,k}$.
>
> The parameter $\eta_{i,k}$ is induced by the hierarchical structure of arms. Here the hierarchical structure is defined as follows: For any agent $P_i$, the $j$th best arm available at the subsequent stage has lower latent utility than the $j$th best arm available at the current stage, where $j\geq 1$. For the optimization problem in Section 3, the $\eta_{i,k}$ serves as a regularization parameter for the uncertainty measure $\delta_{i,k}$. In practice, we may choose a large value of $\eta_{i,k}$ if the agents’ competition is tense, as the arms left at subsequent stages are much worse than the arms available at the current stage.
>
>
>
> [Comment 2] The model makes sense and the abstractions taken to make it workable mathematically are not too detached from reality: my main concern is if there was a way to more explicitly expose the competition aspect between agents? This is encoded through the $\pi_i$’s and $s_{i,k}$: I am wondering how different a model where each arm (agent) has a randomly produced weak ordering of arms would be.
>
> Response: Thank you for the question. We answer this question in three parts.
>
> First, our model can address the case that each agent has a randomly produced weak ordering of arms. Specifically, we incorporated the agents’ competition for arms with a higher score into $\pi_{i,k}(s_{i,k},v_j)$, which depends on the score $v_j$. Here each agent $P_i$ has two values on an arm $A_j$—a score $v_j$ and a fit $e_{ij}$, where $v_j$ is known to all agents and $e_{ij}$ is a private value only known to agent $P_i$. By letting $v_j = 0$ and letting the ordering of $e_{ij}$ correspond to the randomly produced weak ordering of arms, we can address the case that each agent has a randomly produced weak ordering of arms. Moreover, our analysis and results in the paper hold for the case that each agent has a randomly produced weak ordering of arms.
>
> Second, our model allows no restriction on arms’ preferences, which can even involve uncertainty. We incorporated the uncertainty of preferences into the state parameter $s_{i,k}$. Since there is no restriction on arms’ preferences, our analysis and results in the paper hold for the case that each arm has a randomly produced ordering of agents.
>
> Third, it is interesting to study other competition aspects between agents given appropriate motivating examples. We shall leave them for future studies.
>
>
>
> [Comment 3] The fairness results are interesting, but I might have chosen a different definition of envy for this setting: restricting the envy to be ‘per turn’ simplified the analysis but begs the question: If an arm is passed over by its favourite agent and then pulled by less desirable one over two rounds, should it not have envy? Moreover, the arms might not know exactly what the strategy of the agents is. Perhaps it would be more natural to define envy ex-post.
>
> Response: Thank you for the question. We answer this question in two parts.
>
> First, our definition of envy in Section 4 is for ‘per arm’ and ‘per stage.’ That is, an arm has justified envy if it was passed over by its favorite agent at any stage $k\in[K]$. This definition answer the question that if an arm is passed over by its favorite agent and then pulled by a less desirable one over two rounds, the arm indeed has envy, according to the definition of envy in Section 4.
>
> Second, ex-post envy would be an interesting topic. However, it requires a very different set of technical theories and analysis, and we shall leave them for future studies.

---

### Decision · Program_Chairs · 2021-09-27

**Decision:**

Accept (Poster)

**Comment:**

Reviewers were supportive of the present submission, appreciating its approach to a general problem faced in modern two-sided markets.  Reviewers supported the model at a high level, and appreciated the cleanliness of the results proved about that model.  Still, there were concerns raised by some reviewers about the definition of and exposition surrounding fairness (auj5, GLYK) and its application to particular settings (e.g., can preferences of the arms change, and so on; see reviews).  Some responses to the reviewers' questions were not satisfactory, for example, when asked about the complexity of their results, the authors responded with:

-- "We adopted the use of NP-completeness by following the literature; see, e.g., Section 2 of Milgrom (2017, Discovering prices. Columbia University Press). However, for clarity, we will change the writing in lines (155, 181) from NP-completeness to computational intractable."

That citation is a 200+ page general book, and referring to Section 2 of it does not answer the reviewer's question.  Other examples abound.  Still, the rebuttal did address many of the reviewers' concerns.  Should this paper be accepted, this AC would very much appreciate the authors spending substantial time with the reviewers' comments (especially auj5, zGHc, and GLYK) before finalizing their work.